# Dopamine receptor 1 neurons in the dorsal striatum regulate food anticipatory circadian activity rhythms in mice

Christian M Gallardo[1†], Martin Darvas[2†], Mia Oviatt[1], Chris H Chang[3], Mateusz Michalik[4], Timothy F Huddy[5], Emily E Meyer[3], Scott A Shuster[1], Antonio Aguayo[5], Elizabeth M Hill[5], Karun Kiani[3], Jonathan Ikpeazu[1], Johan S Martinez[1], Mari Purpura[3], Andrea N Smit[4], Danica F Patton[4‡], Ralph E Mistlberger[4], Richard D Palmiter[6], Andrew D Steele[1,5]*

[1]Division of Biology, California Institute of Technology, Pasadena, United States; [2]Department of Pathology, University of Washington, Seattle, United States; [3]W M Keck Science Department, Claremont McKenna, Pitzer and Scripps Colleges, Claremont, United States; [4]Department of Psychology, Simon Fraser University, Burnaby, Canada; [5]Biological Sciences Department, California State Polytechnic University Pomona, Pomona, United States; [6]Department of Biochemistry, Howard Hughes Medical Institute, University of Washington, Seattle, United States

*For correspondence: adsteele@csupomona.edu

†These authors contributed equally to this work

Present address: ‡Department of Biology, Stanford University, Palo Alto, United States

**Abstract** Daily rhythms of food anticipatory activity (FAA) are regulated independently of the suprachiasmatic nucleus, which mediates entrainment of rhythms to light, but the neural circuits that establish FAA remain elusive. In this study, we show that mice lacking the dopamine D1 receptor (D1R KO mice) manifest greatly reduced FAA, whereas mice lacking the dopamine D2 receptor have normal FAA. To determine where dopamine exerts its effect, we limited expression of dopamine signaling to the dorsal striatum of dopamine-deficient mice; these mice developed FAA. Within the dorsal striatum, the daily rhythm of clock gene *period2* expression was markedly suppressed in D1R KO mice. Pharmacological activation of D1R at the same time daily was sufficient to establish anticipatory activity in wild-type mice. These results demonstrate that dopamine signaling to D1R-expressing neurons in the dorsal striatum plays an important role in manifestation of FAA, possibly by synchronizing circadian oscillators that modulate motivational processes and behavioral output.

## Introduction

Circadian (~24 hr) rhythms of behavior and physiology are regulated by a distributed system of cell-autonomous circadian oscillators located in the brain and in most peripheral organs and tissues (*Bell-Pedersen et al., 2005*; *Mohawk et al., 2012*). In mammals, a population of coupled circadian clock cells in the hypothalamic suprachiasmatic nuclei (SCN) function as a master pacemaker responsible for coordinating circadian oscillators elsewhere in the brain and other tissues with daily light–dark cycles (*Welsh et al., 2010*). Circadian clocks in many tissues, including multiple brain regions outside of the SCN, can also be entrained by daily cycles of food availability, independently of the SCN pacemaker (*Stokkan et al., 2001*; *Mistlberger, 2011*; *Verwey and Amir, 2011*; *Mohawk et al., 2012*). In rat and mouse, this is readily demonstrated by restricting food access to the middle of the light period, when nocturnal rodents normally eat little and are inactive. This induces a marked shifting of circadian oscillators and organ functions to align with the new daily feeding time, while the SCN remains coupled to the light-dark (LD) cycle. In many species, this is also associated with the emergence of a daily bout of

**eLife digest** If you have ever traveled a long distance by plane, you will likely be familiar with jet lag. This disorientating sensation occurs because our brains have 'internal clocks' that keep track of the day–night cycle and control when we feel most tired or most alert. Flying rapidly from one time zone to another causes this clock to fall out of sync with the local time. It then takes time for the brain's clock to slowly adjust by responding to the levels of light and dark in the new environment.

Humans—and other animals, plants, and even algae—have similar internal clocks, which are used to control behavior and predict events, such as the timing of a meal. These clocks can be set based on previous experiences of when food has been available and can be independent of those that follow the daily cycle of light and dark.

Mice, for example, have internal clocks that make them more active at night and sleep during the day. However, if food is only provided during the day—say, at 2 o'clock in the afternoon—hungry mice will quickly adjust when they are awake in order to get the food as soon it is provided. Also, for a few hours before their new feeding time the mice will tend to jump and move around more; this is known as 'food anticipatory activity'. Researchers have been studying this activity for around 40 years, but the specific regions of the brain and the processes that support these rhythms of feeding behavior remained unknown.

Now, Gallardo et al. have shown that mice need dopamine—a neurotransmitter that is often called the brain's 'feel-good chemical'—to maintain the internal clock that supports food anticipatory activity. Neurotransmitters are chemicals that carry signals between neurons; one neuron releases the chemical, and another detects it using proteins on the neuron's surface called receptors. Two main types of receptors—called D1 receptors and D2 receptors—detect dopamine. Gallardo et al. found that D1 receptors are important for maintaining feeding-related daily rhythms, but that D2 receptors are not. Additionally, dopamine only needs to be produced in a region of the brain called the dorsal striatum for food anticipatory activity to occur. This suggests that only D1 receptors in this region influence this activity, though there are many other regions of the brain that contain these receptors.

The next challenge is to unravel the neural circuits that control food anticipation behavior. For example, what 'tells' the neurons in the dorsal striatum that an animal is hungry? Which of the D1 receptor expressing neurons relay the information about the timing of food anticipatory behavior and to where? Also, if a similar clock operates in humans, testing to see if it is misregulated in people with eating disorders could help us to better understand these conditions.

locomotor activity that anticipates meal time by 1–3 hr (*Boulos and Terman, 1980*; *Stephan, 2002*; *Mistlberger, 2011*). Remarkably, this so-called food anticipatory activity (FAA) exhibits formal properties of circadian clock control, yet persists robustly after removal of the SCN (*Stephan et al., 1979*; *Boulos and Terman, 1980*; *Marchant and Mistlberger, 1997*). Anticipatory rhythms can also be induced by scheduled daily access to water, salt, palatable foods, an opportunity to mate, and psychostimulant drugs (*Mistlberger and Rusak, 1987*; *Mistlberger, 1994*; *Kosobud et al., 1998*; *Iijima et al., 2002*; *Mendoza et al., 2005*; *Honma and Honma, 2009*; *Webb et al., 2009*; *Hsu et al., 2010a*, *2010b*; *Gallardo et al., 2012*; *Jansen et al., 2012*; *Landry et al., 2012*; *Mohawk et al., 2013*). The ability of animals to coordinate activity and physiology with access to critical resources is obviously adaptive. Similar circadian processes in humans could create daily windows of vulnerability to drug seeking, overeating, and other addictive behaviors.

A major knowledge gap in circadian neurobiology is the location of circadian oscillators that generate food (and other reward) anticipatory circadian rhythms. The stimuli and neural pathways that entrain these oscillators also remain unspecified. Conventional lesion experiments have ruled out a number of brain regions as the site of circadian oscillators necessary for FAA (*Mistlberger, 1994*, *2011*; *Davidson, 2009*). This includes the dorsomedial nucleus of the hypothalamus, an area that may nonetheless participate in the expression of daytime FAA by inhibiting output from the SCN pacemaker that normally opposes activity and promotes rest during the day in nocturnal rodents (*Acosta-Galvan et al., 2011*; *Landry et al., 2011*). Induction of anticipatory rhythms by a range of

ingestive and non-ingestive reward stimuli suggest that activation of reward circuits in the brain may play a role, perhaps as a final common entrainment pathway. This is supported by evidence that circadian clock genes exhibit daily rhythms of expression in components of the reward system, including the dorsal striatum and nucleus accumbens (*Wakamatsu et al., 2001*; *Angeles-Castellanos et al., 2007*; *Verwey and Amir, 2011*). These rhythms are inverted by daytime feeding, and, in the dorsal striatum, can be induced or reset by dopaminergic stimuli (*Iijima et al., 2002*; *Hood et al., 2010*; *Natsubori et al., 2013*, *2014*). Importantly, FAA rhythms can be shifted by a single injection of a dopamine D2 receptor agonist (*Smit et al., 2013*) and can be attenuated by dopamine D1 and D2 antagonists (*Liu et al., 2012*). In addition, genetic deletion of factors that activate dopamine transmission (e.g., ghrelin receptors) can attenuate FAA (*Blum et al., 2009*; *LeSauter et al., 2009*; *Lamont et al., 2012*), but see (*Szentirmai et al., 2010*; *Gunapala et al., 2011*; *Patton et al., 2013*) while deletion of factors that suppress DA transmission (e.g., leptin and 5HT1c receptors) can increase FAA (*Mistlberger and Marchant, 1999*; *Hsu et al., 2010c*; *Ribeiro et al., 2011*). Lesion experiments rule out the nucleus accumbens as the site of dopamine receptors that might be necessary for FAA (*Mistlberger and Mumby, 1992*), but comparable experiments of the dorsal striatum have not been reported.

In this study, we used dopamine-deficient mice and dopamine receptor knockout mice to show that food-entrained circadian rhythms are markedly attenuated in the absence of D1 receptors but are spared in mice lacking D2 receptors and in mice that express dopamine only in the dorsal striatum. We further show that a D1 receptor agonist administered once daily in the light period can induce an anticipatory rhythm. Analyses of total daily activity, FAA as a proportion of daily activity, body temperature, and striatal clock gene expression suggest that the FAA phenotype of D1R KO mice may involve impaired synchronization of food-entrainable oscillators combined with an alteration in the strategy for maintaining metabolic homeostasis, favoring reduced total daily activity over nocturnal hypothermia.

## Results

### Absence of the dopamine D2 receptor does not impair FAA

To determine the significance of the dopamine system in mediating FAA, we tested mice lacking either dopamine D1 receptors (D1R KO mice) or dopamine D2R receptors (D2R KO mice). First, we investigated whether mice lacking D2R (*Kelly et al., 1997*) showed impairments in FAA. We measured the baseline home-cage behavior using automated video-based behavior analysis (*Steele et al., 2007*) of D2R KO mice and wild-type (WT) littermates in their home-cage environment. Prior to any dietary intervention ('day -7'), both groups of mice demonstrated normal nocturnal activity waveforms with no significant differences in any 1-hr bin and no increase in activity in the hours before Zeitgeber Time (ZT) 8 (on a 13:11 LD cycle, ZT 12 is lights-off by convention) (*Figure 1A*). Next, we tested their ability to time a daily 60% CR meal fed at ZT 8 every day for 28 days, recording their behavior weekly. Data were normalized by dividing the amount of high activity behavior in each hour by the total seconds of activity over the 24 hr video recording to express a fraction of high activity per hour. Measurements taken at 14, 21, and 28 days of CR revealed that both WT and D2R KO mice had a large increase in high activity (hanging, jumping, walking, and rearing) behaviors during the 3 hr preceding feeding time (*Figure 1B–D*). In both D2R KO and WT mice fed 60% CR daily, we observed similar acquisition and maintenance of FAA defined as normalized high activity in the 3 hr before feeding (ZT 5–8) (*Figure 1E*). As expected, mice of either genotype with ad libitum (AL) access to food showed very little high activity behavior in the hours preceding scheduled feeding when they were given an additional food pellet as a control for handling and disturbance (*Figure 1F*). These results suggest that D2R is not necessary for mediating FAA on a 60% CR meal.

### Absence of the dopamine D1 receptor attenuates FAA

We also examined mice deficient in D1R (*Drago et al., 1994*). The normalized activity waveform of D1R WT and D1R KO mice prior to any dietary intervention on day -7 revealed strong nocturnal activity in D1R KO and control mice, with D1R KO mice showing slightly increased nocturnal activity compared to WT controls (*Figure 2A*, *Figure 2—figure supplement 1*). Summation of time of high activity behaviors in the 11 hr of dark yielded a median value of 93.5 min (min) for D1R WT mice (n = 16) compared with a value of 173.8 min for D1R KO (n = 19); this difference was statistically significant (p = 0.045, Mann–Whitney Test). In comparison, activity values during the lights-on period were not

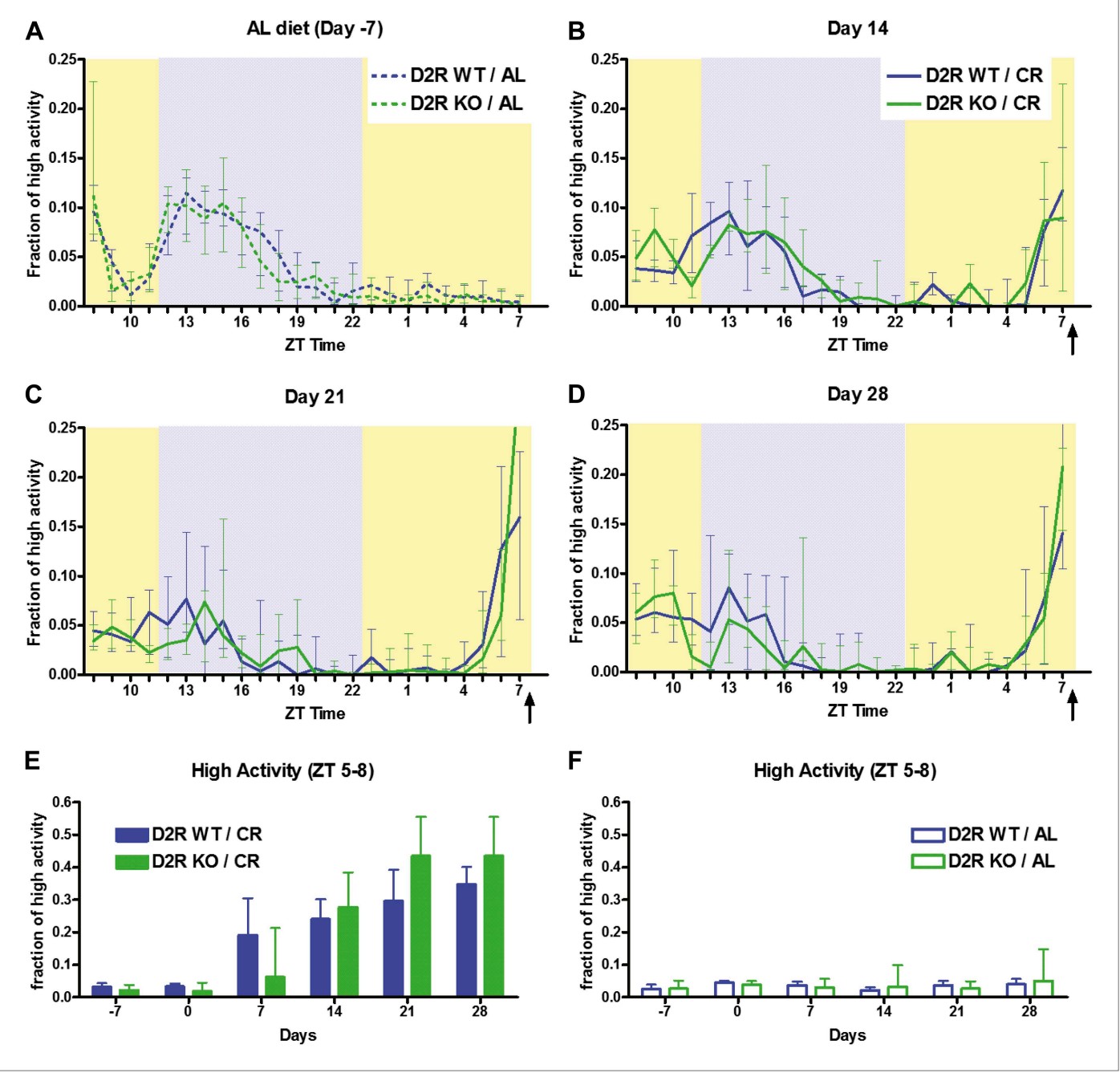

**Figure 1**. Activity of D2R KO mice and WT mice on 60% CR. (**A**) The fraction of all recorded frames within each 1-hr bin on day -7 when the mice were walking, hanging, jumping, or rearing. All mice were still on an ad libitum diet. (**B**, **C**, **D**) The fraction of high activity frames for D2R WT (n = 12) and KO (n = 8) mice in each 1-hr bin on days 14, 21, and 28 of CR. Arrows indicate the bin in which the calorie restricted meal was delivered (ZT 8). Shaded boxes represent lights-off and yellow boxes indicated lights on. (**E**) The fraction of high activity in the 3 hr before feeding time (ZT 5–8) on days -7, 0, 7, 14, 21, and 28 of the study for mice on CR diets. (**F**) The fraction of high activity in the 3 hr before feeding time (ZT 5–8) for mice on ad libitum diets. There were no significant differences (Mann–Whitney) in fraction of high activity between D2R WT and KO mice. Median data are plotted with error bars indicating interquartile ranges.

significantly different (median values of 44.2 and 51.3 min were observed for D1R WT and KO, respectively; p = 0.46) These results suggest that the D1R KO mice do not exhibit locomotor defects preventing nocturnal activity.

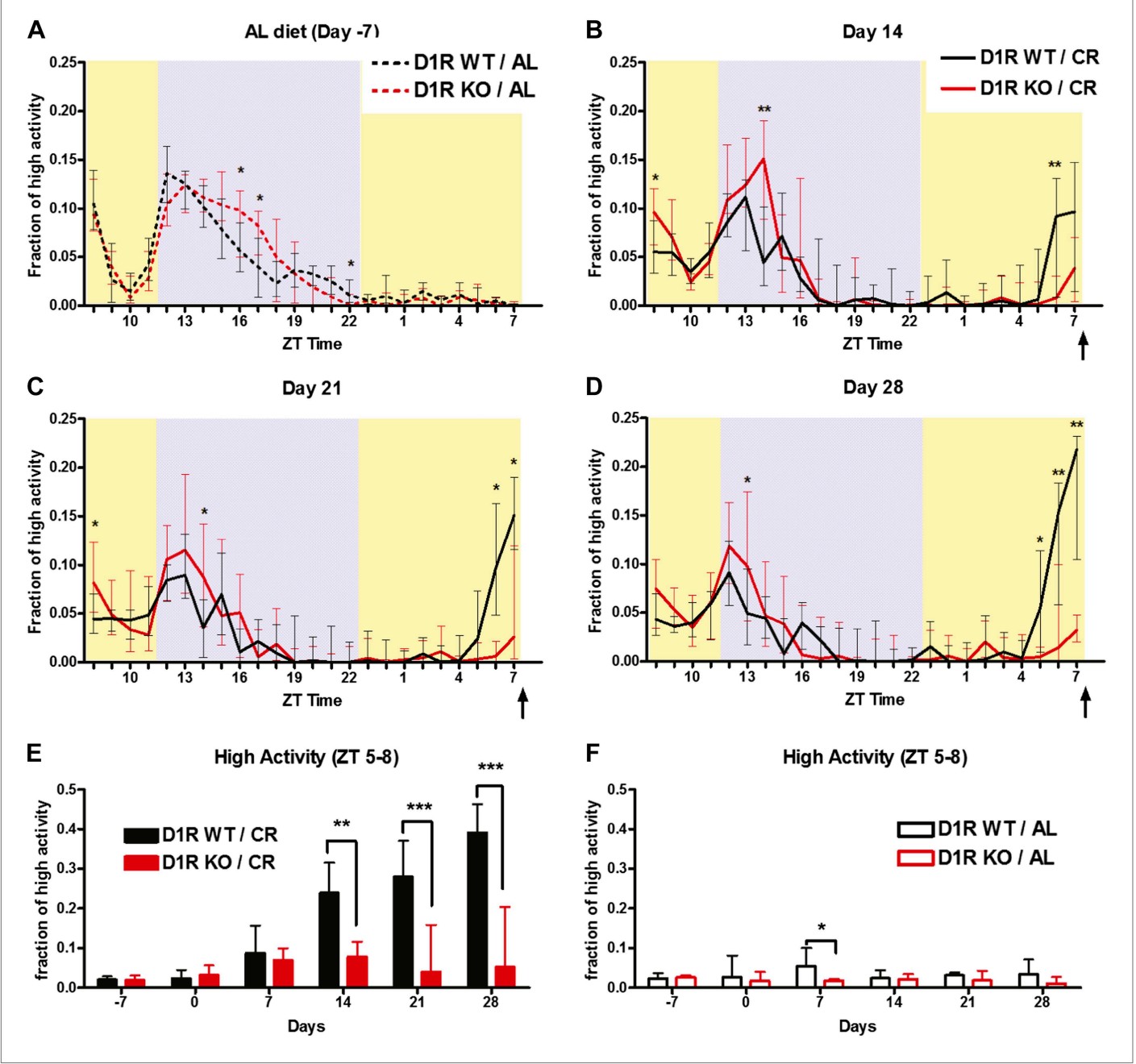

**Figure 2**. Activity of D1R KO (n = 18) mice and WT (n = 16) mice on 60% CR. (**A**) The fraction of high activity within each 1 hr bin on day -7 during which all mice were on an ad libitum diet. (**B**) The fraction of high activity on day 14, (**C**) day 21, and (**D**) day 28 of CR. Arrows indicate the bin in which the meal was delivered (ZT 8). Shaded boxes represent lights-off and yellow boxes indicate lights on. (**E**) Summed normalized high activity in the 3 hr before feeding (ZT 5–8) for days -7, 0, 7, 14, 21, and 28 of the mice on CR diets. (**F**) Summed normalized high activity in the 3 hr before feeding (ZT 5–8) for the mice on AL feeding schedules. Bars show medians and interquartile ranges. The statistical test used was Mann–Whitney, where * indicates p < 0.05, ** indicates p < 0.01, and *** indicates p < 0.001.

The following figure supplements are available for figure 2:

**Figure supplement 1**. High activity data for D1R knockout mice in seconds (median +/- SEM).

**Figure supplement 2**. Individual mouse normalized high activity data from n = 6, WT and mice on day 0 and day 21 of 60% CR diet.

**Figure supplement 3**. Individual mouse normalized high activity data from n = 6, D1R KO mice on day 0 and day 21 of 60% CR diet.

When placed on a timed 60% CR feeding schedule with a ZT 8 meal time, D1R WT mice showed notable increase in activity preceding scheduled meal time by 14 days of CR while D1R KO mice showed only a small increase in activity (*Figure 2B*). On day 21 of CR, WT control mice exhibited a significant ($p < 0.05$) increase in high activity compared to D1R KO mice in ZT 6 and 7 (*Figure 2C*). This was observed again on day 28 with WT mice showing a significantly higher ($p < 0.01$) fraction of high activity in each hourly bin from ZT 5–7 (*Figure 2D*). Summation of the amount of normalized high activity in the 3 hr preceding scheduled meal access revealed that D1R KO mice had a stable and significant impairment in FAA ($p < 0.01$ at day 14, $p < 0.001$ at days 21 and 28) from day 14 of scheduled feeding onwards (*Figure 2E*). As expected, both D1R KO and WT mice with AL access to food showed very little normalized high activity behaviors in ZT 5–8 with the exception of day 7 when D1R WT control mice showed a small increase in activity relative to D1R KO mice (*Figure 2F*). We also plotted the median time (in seconds) of high activity in D1R KO and WT mice over the course of 28 days of CR and observed that D1R KO mice had reduced overall activity on a 60% CR diet after day 14 (*Figure 2—figure supplement 1*). Data from representative D1R WT and KO mice on days 0 and 21 are presented in *Figure 2—figure supplements 2 and 3*, respectively.

To confirm these findings using conventional methods for long-term continuous recording of circadian activity rhythms in mice, D1R KO (n = 4) and WT (n = 12) mice were housed individually in plastic cages with horizontal running discs, within isolation cabinets with motion sensors and controlled lighting (LD 12:12, ~70 lux), temperature (22 ± 2°C range), and humidity (50%).

During AL food access, activity measured by running discs and motion sensors was nearly indistinguishable in amount and timing (% nocturnality) in the KO and WT groups (*Figure 3A,D*). When food (powdered chow mixed with corn oil 20% by weight) was gradually restricted to a 4-hr daily meal for 32 days, beginning at ZT 6 (6 hr after lights-on), WT mice exhibited a bout of disc running that began ~2 hr prior to meal time and increased monotonically to a peak at meal time within a few days (*Figure 3B*). Total daily running did not change. Relative to WT mice, D1R KO mice exhibited significantly less total daily running during food restriction, and a marked reduction in FAA, expressed both as 2-hr total counts and as a ratio relative to activity during the rest of the day, excluding meal

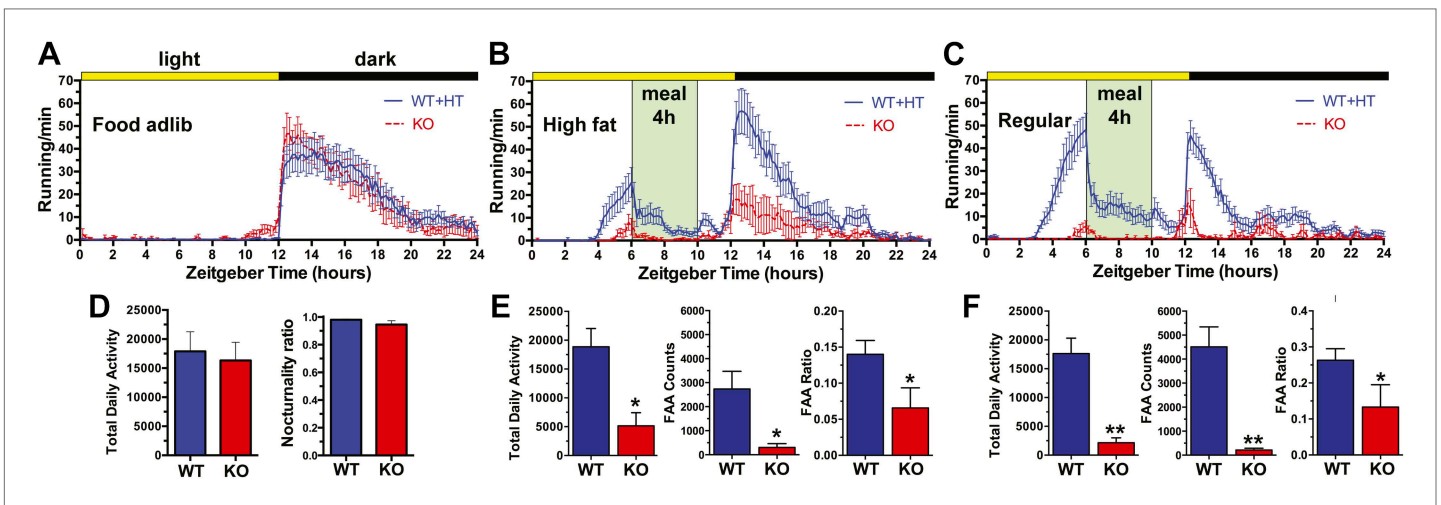

**Figure 3**. Disc-running activity of D1R KO (n = 4, red curves and bars) and WT/HT mice (n = 12, blue curves and bars) during ad-lib and temporal food restriction schedules. (**A**) Group mean (±SEM) waveforms of activity in 10 min bins during ad-lib food access. Data from each mouse are averages of the last 7 days prior to restricted feeding (red dashed lines and bars). Lights-on (ZT 0–12) is indicated by the yellow bar. (**B**) Group mean waveforms of activity during restricted feeding (4 hr daily access to a moderately high fat diet). (**C**) Group mean waveforms of activity during restricted feeding regular chow. (**D**) Total daily activity and nocturnality ratios of WT and KO mice during ad-lib food access. (**E**) Total daily activity, FAA (2-hr pre-meal) counts, and FAA ratios (2-hr pre-meal counts divided by activity during lights-off) during moderately high fat chow schedule. (**F**) The same metrics as panel **E**, during regular chow schedule. * denotes significant difference between WT and KO, $p < 0.05$, 1-tailed. **denotes significant difference, $p < 0.001$.

The following figure supplement is available for figure 3:

**Figure supplement 1**. Actogram data for representative D1R knockout and control mice.

time (p < 0.05; *Figure 3E*). Equivalent effects were evident in activity measured by motion sensors (not shown). KO mice weighed significantly less than WT mice prior to restricted feeding (15.2 ± 2.6 g vs 23.3 ± 2.3 g, p < 0.001) and showed equivalent changes in weight across the 32 days of restricted feeding, losing ~15% body weight over the first week, regaining this by the end of the second week, and remaining stable until day 32 (weight as a percent of baseline = 96 ± 3% vs 105 ± 7% in WT vs KO, respectively, p = 0.22).

The diet of powdered chow mixed with corn oil was used to encourage food intake during the limited time of availability. To determine if the palatability or caloric density of the food influences the magnitude of FAA differentially in KO and WT mice, the diet was changed to powdered chow mixed with water. The mice were fed ad-lib for 22 days and were then restricted to a 4-hr daily meal for 74 days. KO mice continued to show significantly less total activity (p < 0.01) and less FAA (p < 0.01) than WT mice. FAA ratios in both groups were increased relative to the ratios evident when fed high fat chow, but remained significantly lower in KO mice (p < 0.05). In only one KO mice were the FAA counts and the FAA ratios significantly increased during the last 10 days of restricted feeding compared to the last 10 days of ad-lib food access (counts: paired t(9) = 6.52, p < 0.001; ratios t(9) = 5.22, p = 0.005), whereas the counts and ratios were significantly increased in all of the WT mice at p < 0.00001. Differences between KO and WT mice were stable across 74 days on this restricted feeding schedule. Body weights remained stable at ~90% of baseline in both groups. Data from representative D1R KO and WT mice are shown in *Figure 3—figure supplement 1*.

## D1R KO mice have greatly attenuated FAA for more palatable diets

As D1R KO mice show low body weight on standard diets and have a decreased interest in feeding (*Drago et al., 1994*), one possible explanation for their lack of FAA on a 60% CR diet consisting of standard chow (5001 rodent chow; LabDiet) is that it was insufficiently palatable to induce wakefulness that is a prerequisite for FAA. Based on our prior work demonstrating that fatty foods are more potent inducers of FAA in mice than sugary foods (*Hsu et al., 2010b*; *Gallardo et al., 2012*), we tested whether D1R KO mice would show FAA for a 60% CR diet consisting of a more palatable, fat-rich diet. We fed D1R KO and WT controls a 60% CR diet of 'breeder' chow (5015 mouse diet; LabDiet), which has 25.3% calories from fat (5001 rodent chow has 13.5% calories from fat) at ZT 8 daily. D1R KO mice failed to show FAA for breeder chow, exerting only 4–8% of their total daily high activity behaviors in the 3 hr preceding scheduled meal time (*Figure 4A,C*). Interestingly, although WT mice showed FAA for breeder chow, it was attenuated compared to that observed with standard chow, as WT mice redistributed only ~20% of high activity behaviors to the 3 hr preceding feeding (*Figure 4A,C*) compared to 30–40% on standard chow (*Figure 2E*). We also tested an even higher fat content chow, rodent 'high fat diet' (HFD), in which 60% of the calories come from fat. A 60% CR diet of HFD fed once daily at ZT 8 failed to induce FAA in D1R KO mice and induced a very modest FAA in WT controls, which allocated only about 10% of total high activity behaviors to the 3 hr preceding meal time (*Figure 4B,D*). From these experiments, we concluded that D1R KO mice do not fail to anticipate scheduled meal time due to a lack of palatability of the CR food source.

## D1R KO mice have normal increase in fasting-induced activity

Because D1R KO mice failed to anticipate a variety of 60% CR meals fed once daily, we assessed whether this defect was due to an inability to cope with reduced calories. To address this issue, we employed an alternative assay for relating hunger status to activity levels by acutely fasting D1R KO and control mice. In previous studies of the home-cage behavioral response to acute fasting in WT C57BL/6J mice, mice increased their high-activity behaviors two-fold upon acute fasting as compared to AL food access (*Gallardo et al., 2014*). We measured the total high-activity behavior of D1R KO and WT controls over a 3-day period. During days 1 and 2, all mice had AL access to standard chow. As expected, D1R WT and D1R KO mice showed similar median levels of high activity behaviors over 3 days of AL diet (*Figure 5A*). On the third day, mice were either maintained on an AL diet or had their food removed entirely ('fasted'). D1R KO mice showed a strong increase in high-activity behaviors upon acute food deprivation, increasing their activity to a greater extent than fasted WT mice, with mean values of 270.5 min and 376.9 min of high activity, respectively (*Figure 5B*). To normalize these data, we plotted the ratio of activity on fasting day 3 over that of AL day 2 (*Figure 5C*). This ratio is expected to be around one for mice that were not food deprived, in that the activity of an individual mouse should not vary much from day-to-day. Indeed, the amount of high activity on day 2 vs day 3 was

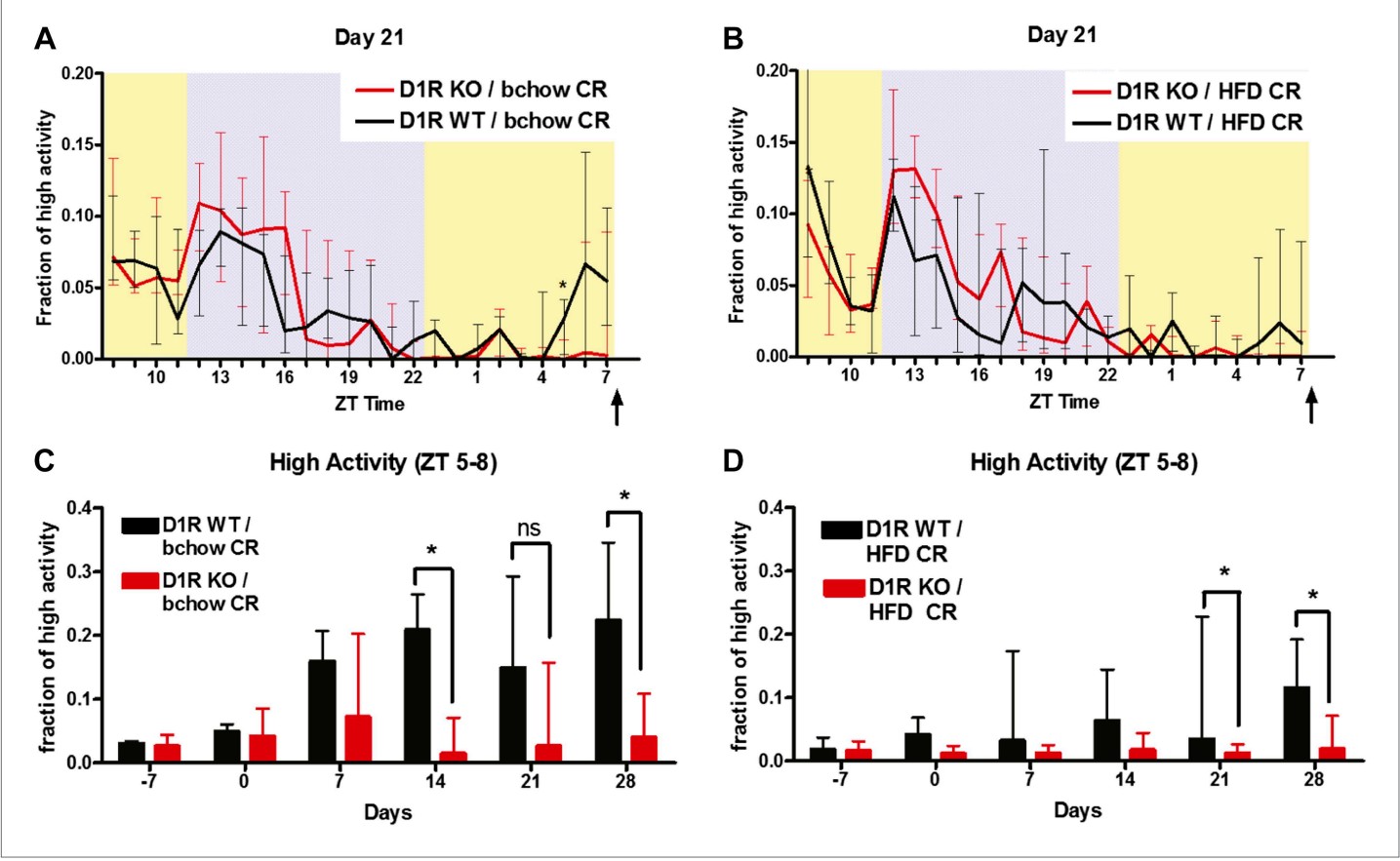

**Figure 4**. Higher fat content diet FAA studies in D1R KO mice. (**A**) Normalized high activity behavior of D1R KO (n = 6) and control (n = 8) mice on day 21 of 60% CR of breeder chow diet. (**B**) Normalized high activity behavior of D1R KO (n = 11) and control (n = 6) mice on day 21 of 60% CR on rodent high fat diet. (**C**) Normalized high activity in the 3 hr preceding scheduled meal time for mice on a diet of 60% CR breeder chow. (**D**) Normalized high activity in the 3 hr preceding scheduled meal time for mice on a diet of 60% CR high fat diet. The statistical test used was Mann–Whitney, where * indicates p < 0.05.

close to one for both D1R WT (0.96) and D1R KO mice (0.92) (*Figure 5C*). However, when food deprived, WT mice had an activity ratio of 1.9 and D1R KO mice had a ratio of 2.5 (both of which were statistically significant when compared within genotype to AL, p < 0.01 Mann–Whitney). Importantly, this assay does not measure food timing, as these mice were naive to any feeding schedule. These results suggest that the D1R KO mice are capable of up-regulating activity in response to acute food deprivation (i.e., hunger), demonstrating an intact circuitry of detecting and responding to fasting and thus ruling out gross metabolic defects.

## Handling cues do not facilitate the expression of FAA in D1R KO mice

Having noted that some D1R KO mice had a small amount of FAA on a 60% CR diet (note the inter-quartile range in *Figure 2B–E*) and that D1R KO mice increased their activity when acutely fasted, we sought to further prompt FAA timing in D1R KO mice. We intentionally cued the D1R KO and control mice to expect food by entering the room in which they were being video recorded and disturbing their cage lids 2 hr in advance () of actual feeding time. We expected that this 'cue' would alert D1R KO mice and perhaps facilitate FAA. We cued D1R WT and KO mice on an AL diet to assay the amount of activity that occurs as the result of a disturbance independent of FAA. We observed a small increase in high activity behaviors lasting only 10 min post-cue in both D1R WT and D1R KO mice; there were no significant differences in the amount of activity induced in D1R WT and KO mice (*Figure 6A*). D1R KO mice on a CR diet did not show a sustained increase in activity after being cued to feeding time; their activity lasted only ~10 min subsequent to the cue (*Figure 6B*). WT control mice on a CR diet showed a large increase in activity induced by the cue, or were already demonstrating FAA at the time

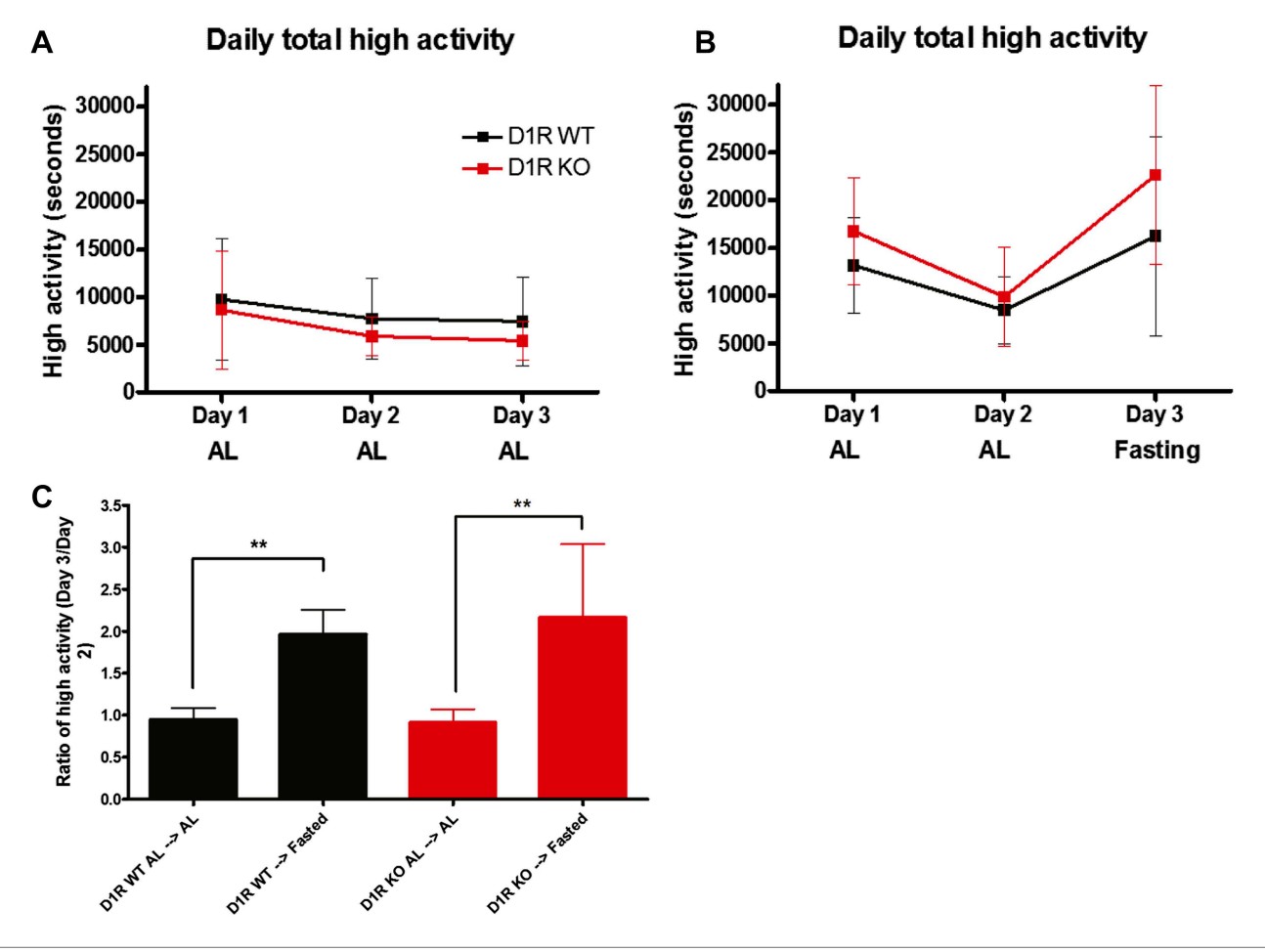

**Figure 5**. Activity of acutely fasted D1R KO and WT mice. (**A**) Total number of seconds of high intensity activity (walking, hanging, jumping, or rearing) for D1R KO (n = 6 KO) and WT (n = 12) mice on 3 consecutive days of ad libitum diet. (**B**) Total number of seconds of high intensity activity for D1R KO (n = 7) and WT (n = 14) mice on 3 consecutive days. On day 1 and day 2 all mice were on an ad libitum diet, but on the third day all mice were deprived of food. (**C**) The ratio of total seconds of high activity on day 3 divided by total seconds of high activity on day 2. Bars show medians and interquartile ranges. The statistical test used was Mann–Whitney, where ** indicates p < 0.01.

of the cue, and sustained their increased activity for the following 2 hr (*Figure 6B*). Summation of the time of high-activity behavior 2 hr prior to feeding demonstrated a significant increase in activity in WT mice when CR compared to AL feeding, but not in D1R KO mice (*Figure 5C*). These results suggest that even cuing the D1R KO mice to the subsequent availability of food did not trigger FAA behaviors.

## Palatable meal anticipation by D1R KO mice on an ad libitum diet

Rodents exhibit FAA in response to a timed palatable mid-day meal even with AL access to regular chow (*Mistlberger and Rusak, 1987*; *Mendoza et al., 2005*). This model does not involve substantial weight loss and mice, in particular, show FAA for fatty meals as opposed to sugary meals (*Hsu et al., 2010b*; *Gallardo et al., 2012*). To gain further insight into the role of the dopaminergic system in FAA, we next studied FAA in response to scheduled palatable meals. We fed a daily meal of rodent HFD at ZT 9 to D1R KO and WT male mice daily for 14 days while they retained AL access to standard chow. The palatable meal corresponded to about 30% of their total daily caloric intake.

D1R KO mice, which often fail to eat from the wire food bin and need to be fed their standard chow on the cage floor, approached the HFD meal placed in their food bin immediately and started consuming it avidly. Both WT and D1R KO mice consumed their HFD meal in less than 1 hr, spending 10%

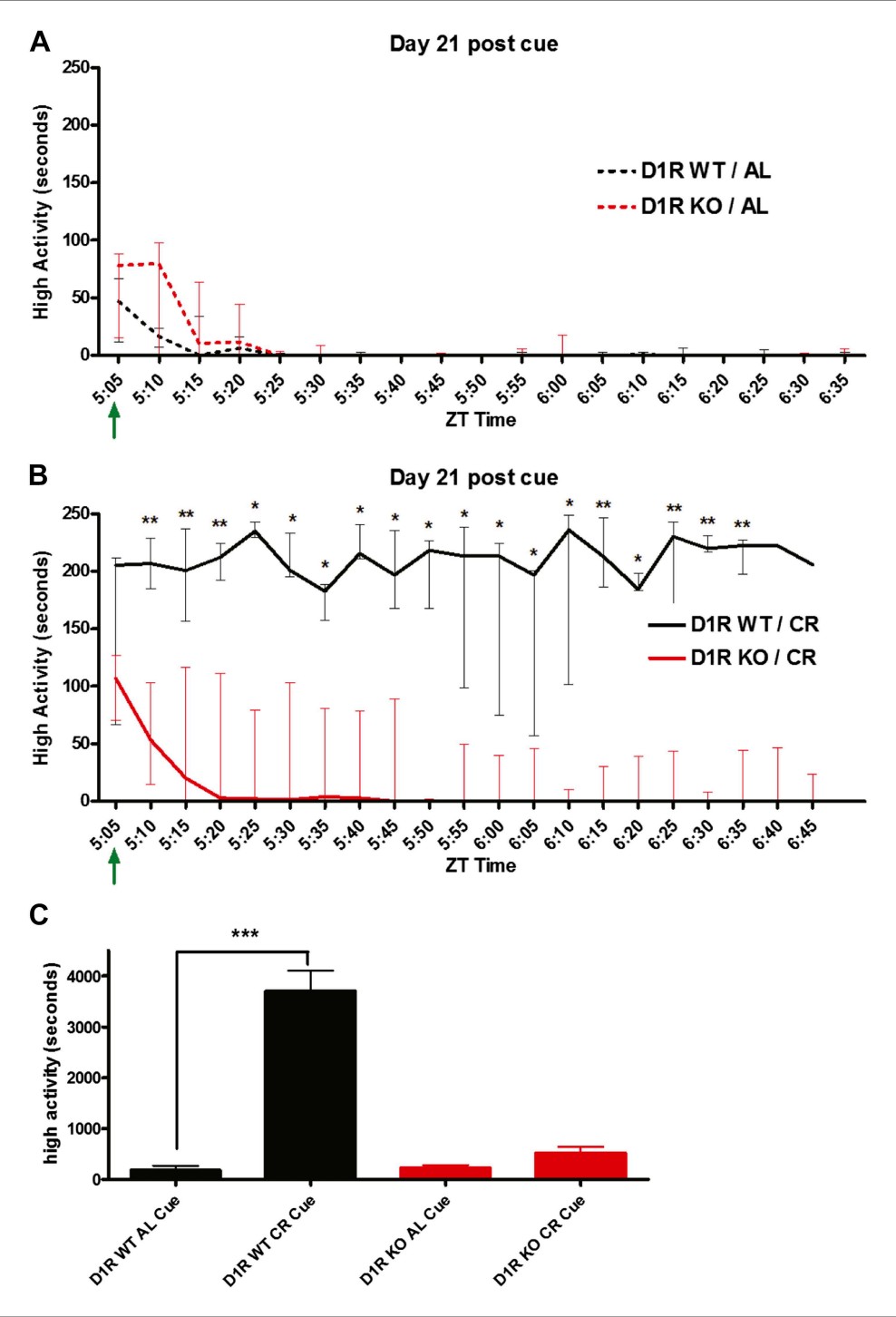

**Figure 6**. Cued handling FAA. (**A**) High activity data (in seconds) in 5 min bins for D1R KO (n = 7) and D1R WT (n = 8) mice that were disturbed 2 hr prior to scheduled feeding. (**B**) High activity data (in seconds) in 5 min bins for timed, calorie restricted D1R KO (n = 6) and D1R WT (n = 7) mice. Mice were disturbed 2 hr prior to feeding. (**C**) Summed high activity data over the cued period. The statistical test used was Mann–Whitney, where * indicates p < 0.05, ** indicates p < 0.01, and *** indicates p < 0.001.

of their food-bin entry time during the hour after palatable-diet feeding (*Figure 7A*). We noted that preprandial food-bin entry, a hallmark of HFD meal entrainment (*Hsu et al., 2010b*), was elevated in D1R KO mice by ZT 8 on the 14th day of scheduled palatable meals (*Figure 7B*). On day 14, both

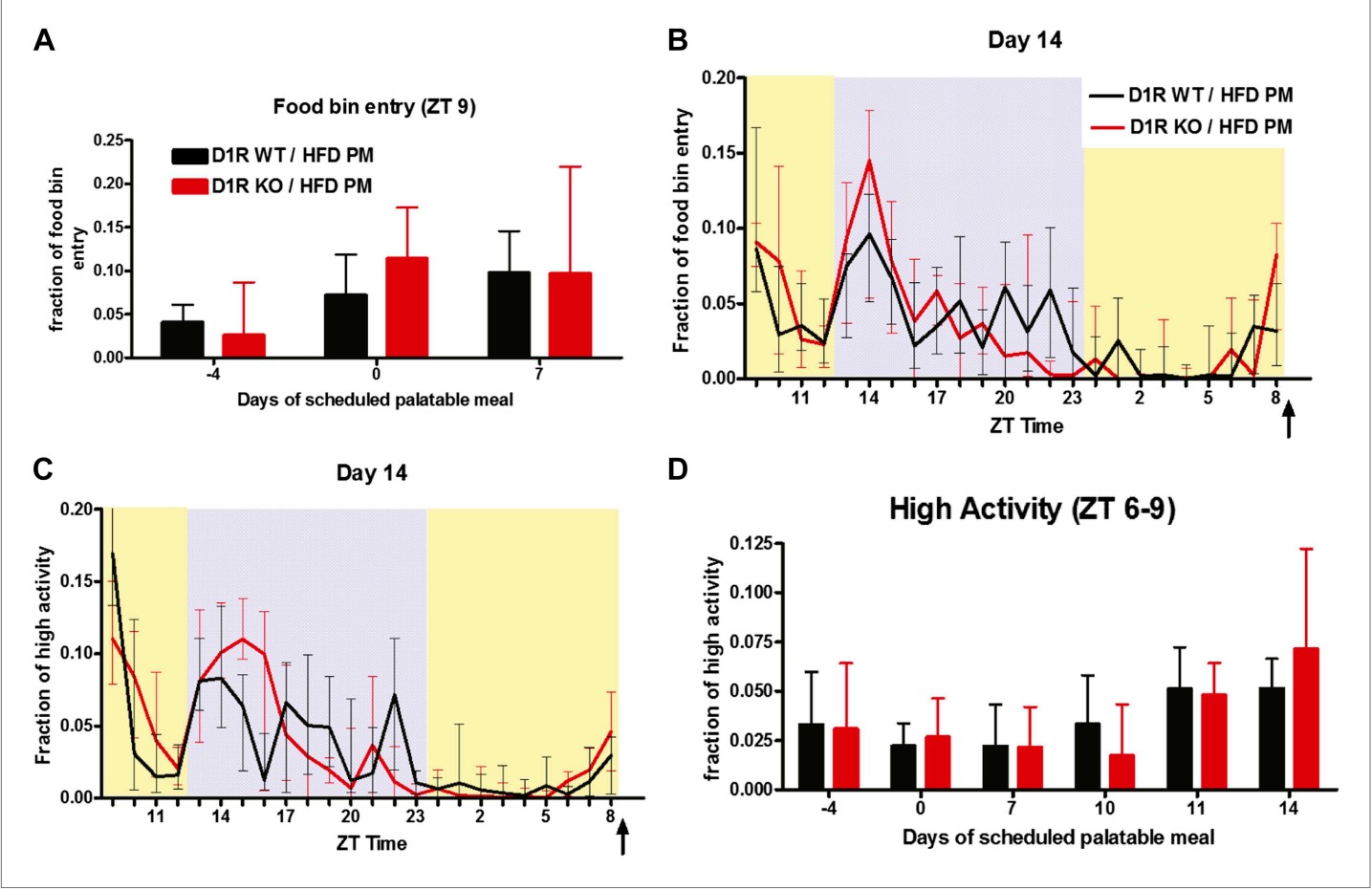

**Figure 7**. Activity of D1R KO (n = 9–11) and WT (n = 14) mice on a palatable meal schedule. (**A**) Fraction of time spent entering the food bin in the hour after feeding (ZT 9) on days 4, 0, and 7. (**B**) The fraction of normalized food bin entry and (**C**) normalized high activity in each 1-hr bin on day 14. (**D**) Sum of normalized high activity in the 3 hr before feeding on days 4, 0, 7, 10, 11, and 14. There were no statistically significant differences between groups, Mann–Whitney. Arrows indicate the bin in which the palatable meal was delivered (ZT 9). Bars show medians and interquartile ranges.

groups of mice exhibited similar activity patterns over the course of the day, with small, but similar, increases in activity preceding their HFD meal (**Figure 7C**). Summing high activity in the 3 hr prior to feeding (ZT 6–9) shows a similar trend of increased fraction of high activity after 10 days of HFD meal treatment for both D1R KO and control mice (**Figure 7D**). Given how poorly D1R KO mice expressed FAA for 60% CR meals, it is surprising that daily HFD meals elicited even modest FAA, leading us to retest FAA for 60% CR meals in the same cohort of mice.

## Pre-treatment with scheduled palatable meals allows D1R KO mice to express FAA transiently

Subsequent to being maintained on a daily HFD meal schedule for 2 weeks, we converted D1R KO and control mice to a standard AL chow diet without palatable meal access for 1 week, days 14–21 (**Figure 8A**). Then, we changed their diet to a daily scheduled 60% CR meal consisting of standard chow at ZT 9 on day 22. On the first day of 60% CR, we did not observe any differences in behavior between D1R KO and control mice as neither group showed FAA, as expected (**Figure 8B**). However, after just 1 week ('day 28') of 60% CR, the D1R KO mice that had previously been treated with HFD showed marked FAA for their daily feeding, with a notable FAA peak that was as high as the night time activity peak (**Figure 8B**). There was no significant difference between the amount of FAA exhibited between D1R KO and WT mice on day 28 (p = 0.734, Mann–Whitney). Remarkably, by the 14th day of 60% CR ('day 35' of the experiment), the D1R KO mice no longer showed FAA (**Figure 8D,G**) and the difference with control mice was highly significant (p = 0.0003, Mann–Whitney). This lack of FAA in

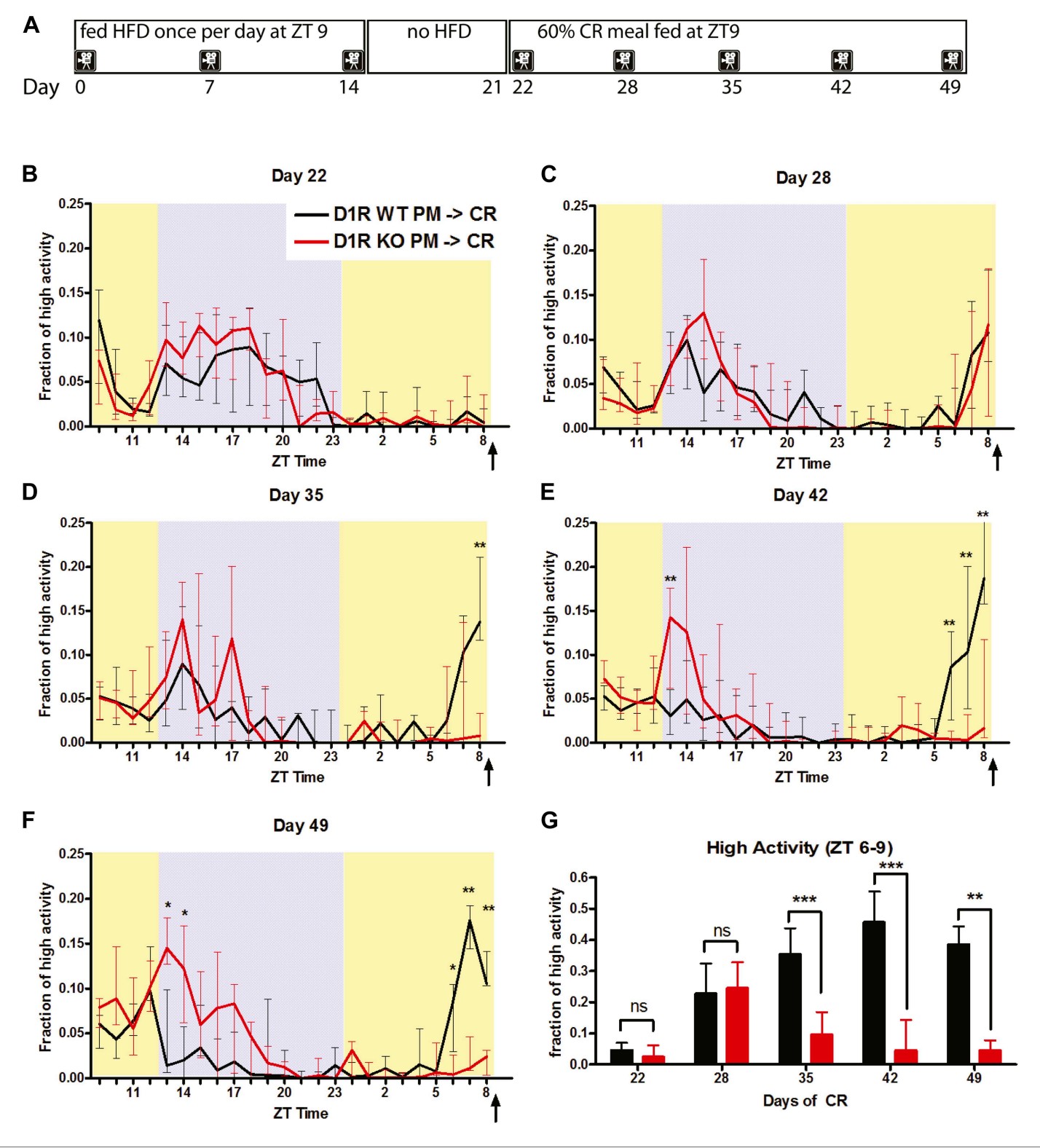

Figure 8. Activity of D1R KO (n = 9) and WT (n = 14) mice on a 60% CR meal pre-treated with 14 days of a palatable meal schedule. (A) A diagram representing the feeding schedule used in this study. (B) The fraction of high activity each 1-hr bin on day 22, (C) day 28, (D) day 35, (E) day 43, and (F) day 49. Arrows indicate the scheduled feeding time. Shaded boxes represent lights-off while yellow represents lights on. (G) The fraction high activity in the 3 hr before feeding time (ZT 6–9) on day 22, 28, 35, 42, 49. Bars show medians and interquartile ranges. The statistical test used was Mann–Whitney, where * indicates p < 0.05, ** indicates p < 0.01, and *** indicates p < 0.001.

D1R KO mice persisted through days 42 and 49, whereas control mice maintained on a CR diet continued to express robust FAA (*Figure 8E,F*). We performed a Kruskal–Wallis test to determine whether there were significant within-genotype variations in this experiment and found that WT mice showed significant differences between FAA observed at day 22 and days 28, 35, 42, and 49, whereas D1R KO mice only showed significant differences in FAA between day 22 and day 28 but not any other day, indicating that the D1R KO mice transiently up-regulate FAA whereas control mice sustain this increase. We conclude that D1R-mediated signaling is not absolutely required for FAA.

## Body temperature entrainment in D1R KO mice on scheduled feeding

Caloric restriction can induce torpor in WT mice, which could mask entrainment of circadian oscillators by food. Visual observations at meal time provided no evidence for torpor in D1R KO mice. To rule out defects in thermoregulation as a reason why D1R KO mice do not show FAA, we placed D1R KO (n = 7) and WT (n = 7) mice at 30°C continuously. After 1 week at 30°C, we tested their behavioral response to 60% CR for the next 4 weeks and did not observe FAA in D1R KO mice and noted a marked reduction in FAA for the WT mice at this elevated temperature compared to ambient (22–24°C) temperature (*Figure 9—figure supplement 1*).

To address body temperature regulation more directly, we implanted miniature radiofrequency transmitters (Minimitter) for continuous recording of body temperature by telemetry in D1R KO (n = 6) and WT (n = 8) mice, which were then maintained on 60% CR feeding at ZT 8. The 24-hr body temperature profiles showed identical waveforms for WT and KO groups before CR schedules were started ('day -3') with temperatures remaining close to 34.5°C during the daytime and increasing to up to ~36°C at night-time (*Figure 9A*). By day 14 of CR, there was a similar redistribution of mean temperature waveforms in both D1R KO and WT mice, starting with a slightly elevated postprandial temperatures and steady temperature declines in the middle of the dark phase (ZT 19) that continues until about 6 hr before expected meal time (*Figure 9B*). Despite similar profiles in temperature waveforms, we did note that D1R KO mice tended to show a slightly higher temperature during the entirety of the dark cycle compared to WT controls during CR (*Figure 9B–D*, *Figure 9—figure supplement 2*). We tested for an effect of genotype and day of measurement on nocturnal temperature, finding that there were no statistically significant differences between D1R WT and KO (genotype accounts for 5.63% of the total variance; F = 1.15; p = 0.3044) but there was a significant effect of day of experiment (interaction accounts for 4.43% of the total variance; F = 4.53; p = 0.0001; 2-way repeated measures ANOVA). However, there was no significant difference in average temperatures between D1R KO and controls during the 3 hr preceding feeding at any point during the experiment (*Figure 9E*). We also examined the fluctuation in temperature, 'the ΔT', in the 3 hr preceding meal time by subtracting the maximum and minimum temperature values within the FAA window (*Figure 9F*). By this metric, D1R KO and WT mice show increasing ΔT following the start of CR feeding schedules with values peaking by day 7 (*Figure 9F*). The D1R KO mice had a similar ΔT to control at all times except for days 14 and 17, where this difference was statistically significant (p < 0.05, day 17). Given that the ΔT is similar again by day 21, we conclude that overall there are subtle differences in temperature entrainment in D1R KO mice but that overall their metabolism is effectively modulated by CR feeding; importantly, they do not fail to show FAA due to entering a torpor state.

## Dopamine signaling solely in the dorsal striatum permits FAA

To determine where in the striatum D1R neurons are signaling FAA, we employed the dopamine-deficient (DD) mouse model. In this system, the rate-limiting enzyme for dopamine production, tyrosine hydroxylase (TH), has a 'lox-stop-lox' cassette within the first intron of the *Th* gene and does not make any dopamine (*Zhou and Palmiter, 1995*; *Zhou et al., 1995*). This lox-stop-lox cassette can be removed by the action of Cre-recombinase, allowing for region specific re-activation of this allele (*Szczypka et al., 1999*, *2001*; *Hnasko et al., 2006*). Cre-recombinase was encoded in canine adenovirus (CAV), which infects neuronal terminals and undergoes retrograde transport. CAV-inoculated mice showed expression of dopamine restricted to neurons that innervate the dorsolateral striatum (*Figure 10A,B*). Quantitative analysis of the TH straining revealed a significant difference in restoration when comparing dorsal vs ventral striatum or anterior commissure (aca), which is always devoid of TH staining (p = 0.018, Kruskal–Wallis test; p = 0.0486 comparing dorsal vs ventral, p = 0.0112 for dorsal vs aca, and p = 0.999 for ventral vs aca, Dunn's post-test for multiple comparisons). To these virally rescued DD mice (DD-VR), or controls (WT mice injected with virus), we presented a 60% CR meal at ZT 9 for

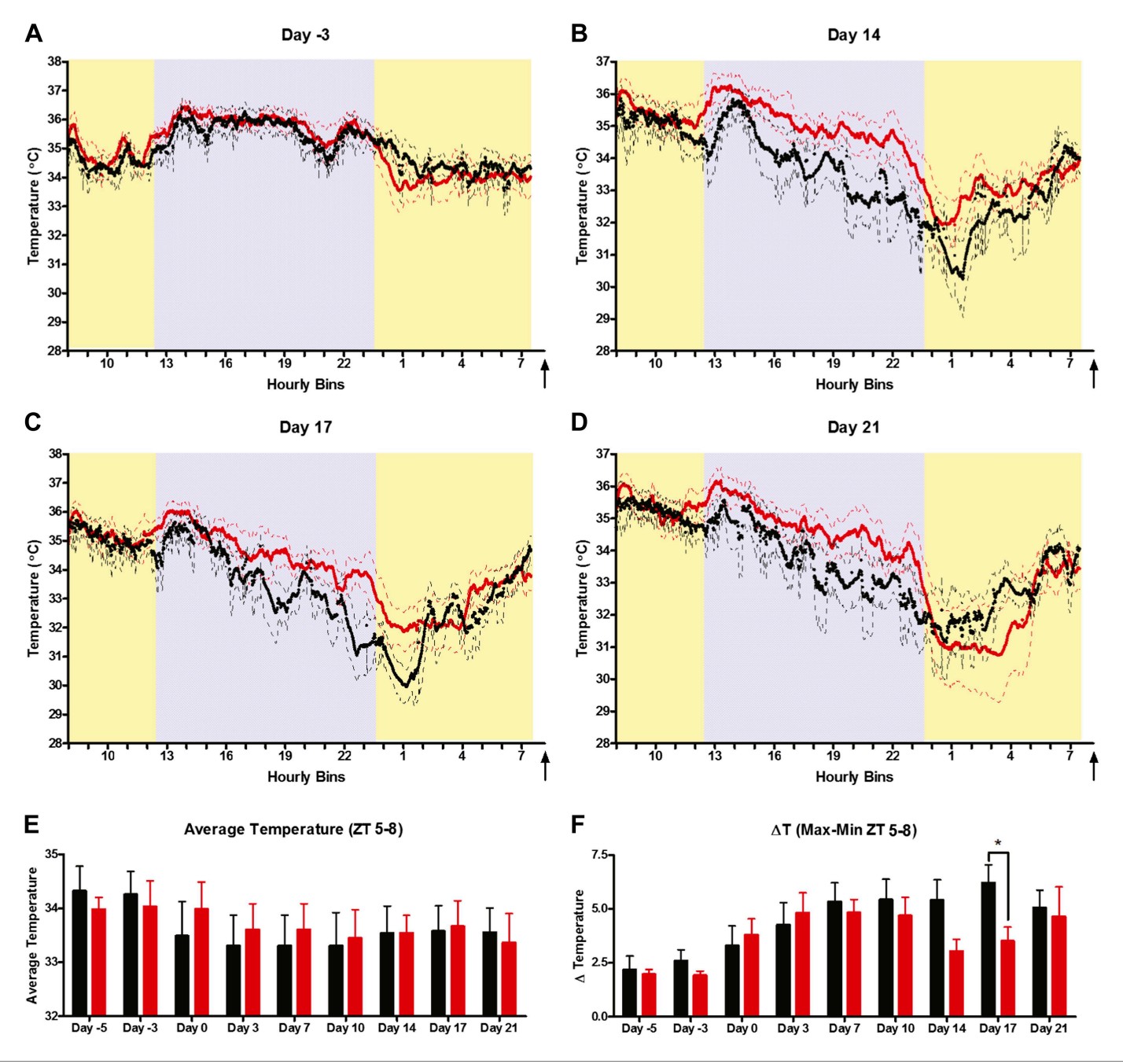

**Figure 9**. Body temperature measurements. (**A**) Mean (± SEM) body temperature of WT and D1R KO mice 3 days prior to initiating CR, (**B**) day 14, (**C**) day 17, and (**D**) day 21 of CR. (**E**) Mean (±SEM) body temperature overall for each day of measurement. (**F**) Mean change in temperature in the 3 hr prior to scheduled feeding. * indicates $p < 0.05$, Mann–Whitney; n = 6 D1R KO and n = 8 WT.

The following figure supplements are available for figure 9:

**Figure supplement 1**. FAA study of D1R KO (n = 7) and WT (n = 7) mice at 30°C.

**Figure supplement 2**. Entrainment of behavior and neuronal activation from D1 agonist injection.

21 days. Initially the activity waveforms of the DD-VR mice showed a trend toward increased activity at night (**Figure 10B**), and they did not show appreciable FAA at the 7-day-time point ($p < 0.05$) (**Figure 10C**). However, by day 14 and onwards they showed a strong FAA response, eventually having almost all of

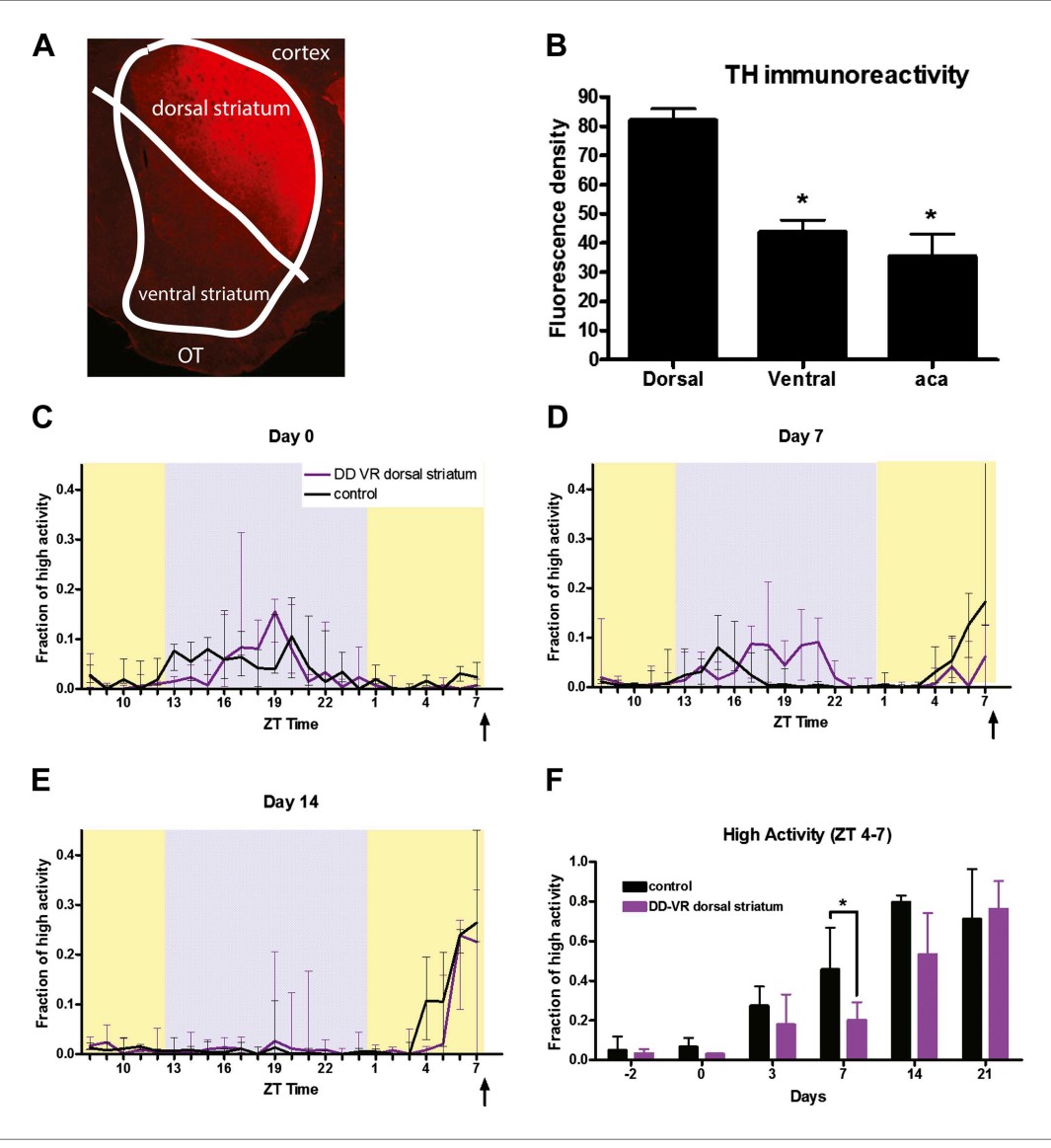

**Figure 10**. Viral restoration of dopamine signaling in the dorsolateral striatum of dopamine-deficient mice.
(**A**) Representative tyrosine hydroxylase staining in a dopamine-deficient dorsolateral viral restoration mouse.
(**B**) Quantitation of tyrosine hydroxylase expression in dopamine deficient mice (n = 5). TH immune-stained striatal
sections from DD-VR mice were analyzed with MacBiophotonics ImageJ software to measure fluorescence
intensities in the dorsal striatum, ventral striatum, and also in the anterior part of the anterior commissure (aca), a
structure that is always devoid of TH staining. For each mouse fluorescence intensity values were divided by the
size of the analyzed area to generate normalized fluorescence values. (**C**) Normalized high activity in control
(normal dopamine levels, n = 4) and dopamine-deficient viral restoration mice (n = 7) on the first day of CR.
(**D**) Normalized high activity on day 7 of CR and (**E**) day 14 of CR. (**F**) Summation of normalized high activity in
the 3 hr preceding meal time over the course of the experiment. * indicates p < 0.05, Mann–Whitney.

their daily activity preceding meal time as did sham surgery controls (*Figure 10D*). From this experiment
we conclude that dopamine signaling in the dorsal striatum is sufficient for acquisition of FAA.

## Dopamine D1 receptor KO attenuates circadian rhythm of Per2 expression in the dorsal striatum

Circadian rhythms in mammals are generated at the single-cell level by autoregulatory transcription–
translation feedback loops involving so-called circadian clock genes and their protein products. A core

loop is comprised of the clock genes *Per1*, *Per2*, *Cry1,* and *Cry2*, which are positively regulated by heterodimers of BMAL1 and CLOCK and negatively regulated by hetero- and homodimers of PER and CRY proteins. Circadian clock genes exhibit 24-hr rhythms of expression in the dorsal striatum (*Harbour et al., 2013*), and these rhythms can be shifted by daytime feeding schedules (*Wakamatsu et al., 2001*) and dopamine agonists (*Hood et al., 2010*). If attenuation of FAA rhythms in D1R KO mice is due to impaired entrainment of circadian oscillators in the dorsal striatum, then daily rhythms of clock gene expression in food-restricted D1R KO mice should be significantly attenuated or differently phased relative to WT mice. WT and KO mice were fed regular chow once daily in the light period (ZT 6–10) for at least 30 days and were then euthanized at 1 of 4 time points (ZT 0, 6, 12, 18) for quantification of *Per2* expression by quantitative, reverse-transcriptase PCR. A 2-way ANOVA revealed a significant effect of time of day ($F_{(4,30)}$ = 5.07, p = 0.0031), genotype ($F_{(1,30)}$ = 10.08, p = 0035), and interaction ($F_{(4,30)}$ = 22.02, p < 0.0001) in *Per2* expression. WT mice exhibited a 24-hr rhythm with peak expression at ZT 12 (lights-off), which is advanced by comparison with the daily rhythm previously reported in AL fed rodents (e.g., *Harbour et al., 2013*). KO mice exhibited a markedly attenuated daily rhythm of striatal *Per2* expression, due primarily to reduced expression at ZT 12 (*Figure 11*). These results permit a substantive conclusion that the daily rhythm of *Per2* expression evident in the dorsal striatum of WT mice anticipating a daily meal is dependent on dopamine signaling at D1 receptors.

## Daily timed pharmacological activation of D1R entrains behavior

We next asked whether deliberate timed activation of D1R neurons is sufficient for behavioral entrainment in the absence of dietary restriction. We injected WT C57BL/6J male mice daily intraperitoneally with D1R agonist SKF-81297 for 12 days (see top panel in *Figure 12*). As controls, we injected additional mice daily with either water as a negative control for IP injection or caffeine as a negative control for activation of the locomotor system, since caffeine induces hyperactivity even in dopamine-deficient mice it is a dopamine-independent pathway (*Joyce and Koob, 1981*; *Kim and Palmiter, 2003*). The immediate response to injection was observed as an increase in activity in caffeine and SKF-81297 mice that was more prolonged than that of water-injected controls (*Figure 12A*). This trend continued through day 12, where water-injected mice showed a habituation to injection but caffeine-injected mice showed increased activity (*Figure 12B*). On day 13, when no injection was performed, leaving all mice undisturbed (no cage changes), the SKF-injected mice showed increased activity compared to both water and caffeine-injected controls during the 3 hr after scheduled injection (*Figure 12C*). We examined mice at days 0, 7, 12, and 13 for evidence of behavioral entrainment to injection. On day 0, there was no increase in activity preceding daily injection, as expected (*Figure 12D*). After 7 days of injection, we observed a trend toward increased activity in SKF-injected mice that was not statistically significant (p = 0.0513, Kruskal–Wallis test) (*Figure 12E*). By days 12 and 13, the fraction of high activity of SKF-injected mice was significantly greater than either that of water- or caffeine-injected mice (day 12, SKF-81297 vs water, SKF 81297 vs caffeine [p < 0.01]; day 13, SKF 81297 vs water [p < 0.01], SKF-81297 vs caffeine [p < 0.01]) (*Figure 12F–H*).

We repeated this experiment, omitting caffeine, and prevented food intake for 4 hr post-injection to control for drug-induced feeding as a potential factor for inducing anticipatory activity in the experiment described above. To that end, mice were injected daily for 14 days with either SKF-81297 (n = 12) or water (n = 12). We observed significant (p < 0.05) behavioral entrainment by the 14th day of injection (*Figure 12—figure supplement 1*). To confirm that the SKF-81297 was activating D1R neurons, we injected n = 2 D1R WT and D1R KO mice with drug and euthanized them 1 hr after injection, processing their brains of c-Fos immunostaining. We noted a marked number of cells staining in the dorsal striatum of WT but not the D1R KO mice (*Figure 12—figure supplement 1*).

From these experiments, we conclude that timed activation of D1R neurons is sufficient to induce moderate behavioral entrainment (about 20% of total high activity behavior in a 3-hr pre-injection window). Moreover, this entrainment can persist in the absence of stimuli, as at the end of day 13 anticipatory activity occurs nearly 46 hr after the last injection of drug.

## Discussion

In rodents, behavioral anticipation of a regularly scheduled feeding time is mediated by circadian oscillators entrainable by feeding. Localization of these oscillators and their entrainment pathways has been an enduring challenge (*Davidson, 2009*; *Mistlberger, 2011*). Recent studies suggest a role for dopamine signaling in the expression and timing of food anticipatory rhythms in mice and rats

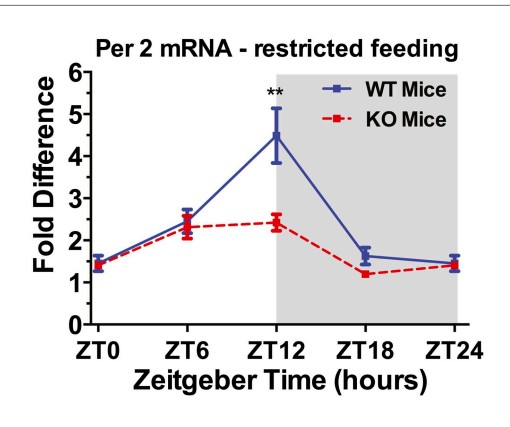

**Figure 11**. *Per2* mRNA expression measured by quantitative reverse-transcriptase PCR at 4 times of day in D1R KO (red dashed curve) and WT mice (blue curve) fed for 4 hr daily at Zeitgeber Time 6. n = 4 mice per group per time point. ANOVA confirms a significant effect of sample time in both groups. **denotes significant difference between KO and WT mice, p < 0.0001.

(*Liu et al., 2012*; *Smit et al., 2013*). In this study, we provide comprehensive evidence supporting an important role for dopamine signaling via D1R in the expression of FAA in CR mice fed in the light period. Using video-based automated behavior analysis, motion sensors, and running discs, we demonstrated that anticipation of a daily meal is markedly attenuated in mice lacking D1R. By contrast, it is normal in mice lacking D2R and in dopamine-deficient mice with local production of dopamine in the dorsal striatum. The reduction in FAA in D1R KO mice is evident both in total counts and when normalized for total daily activity. We propose that attenuated FAA in D1R KO mice may involve three distinct processes; circadian oscillator entrainment, incentive motivation, and metabolic homeostasis.

Circadian clocks regulate the timing of behavior, but can also regulate the peak level of activity by variations in clock amplitude. Mammalian circadian clocks are comprised of populations of coupled circadian oscillators (clock cells) (*Colwell, 2000*; *Mohawk and Takahashi, 2011*). Disruption of coupling within a population of circadian oscillators can flatten aggregate clock output and attenuate rhythm peaks. A circadian oscillator entrainment interpretation of attenuated FAA in D1R KO mice is suggested by two observations. First, in parallel with attenuated FAA rhythms, D1R KO mice also exhibited a marked attenuation of a daily rhythm of clock gene (*Per2*) expression in the dorsal striatum. Second, a daily injection of the D1 agonist SKF-81257 was sufficient to induce a rhythm of anticipatory activity that was synchronized to the injection time and that persisted during a withdrawal day. These results converge on a hypothesis that dopamine signaling via D1R in the dorsal striatum entrains circadian oscillators that regulate the timing and amplitude of FAA. We speculate that a daily rhythm of dopamine release associated with food acquisition serves to coordinate the phase of circadian clock cells in the dorsal striatum, and that genetic deletion of D1R impairs this function, thereby reducing the amplitude of the population rhythm and the magnitude of the activity rhythm driven by these oscillators. In this respect, D1R signaling in food-entrainable oscillators of the dorsal striatum may be analogous to VPAC receptor signaling in the SCN, which is critical for coupling the population of circadian oscillators that constitute this light entrained circadian pacemaker (*Harmar et al., 2002*; *Aton et al., 2005*).

Circadian clocks are thought to directly drive or gate output of neural systems that produce observable daily rhythms of rest and activity. Circadian clocks also generate internal time cues that animals can use to discriminate time of day; for example, to permit daily time-place learning, or time-compensated-sun-compass orientation (*Mistlberger, 1994*). A representation of circadian clock phase, by repeated association with a fixed daily meal time, could acquire salience as an incentive stimulus and generate a daily rhythm of incentive motivation manifest as FAA. Such a model could account for the ability of some animals to anticipate two or more meal times per day (*Biebach et al., 1991*; *Van der Zee et al., 2008*; *Luby et al., 2012*; *Mistlberger et al., 2012*; *Mulder et al., 2013*). It is therefore possible that attenuated FAA in D1R KO mice, which did not express FAA even when aroused 2 hr prior to meal time (*Figure 6*), reflects impaired incentive motivation, due to a failure to attribute incentive salience to representations of circadian phase, due to degradation of the clock signals by striatal oscillator damping, or both.

An unexpected finding was that while a palatable HFD did not improve FAA in CR D1R KO mice, the same diet provided once daily did induce a weak FAA equivalently in KO and WT mice with AL access to regular chow throughout the day. Furthermore, upon transfer to the CR schedule, the KO mice showed a transient FAA that dissipated over the course of the first week. These observations suggest that enhanced palatability can compensate for deficits in oscillator entrainment or incentive

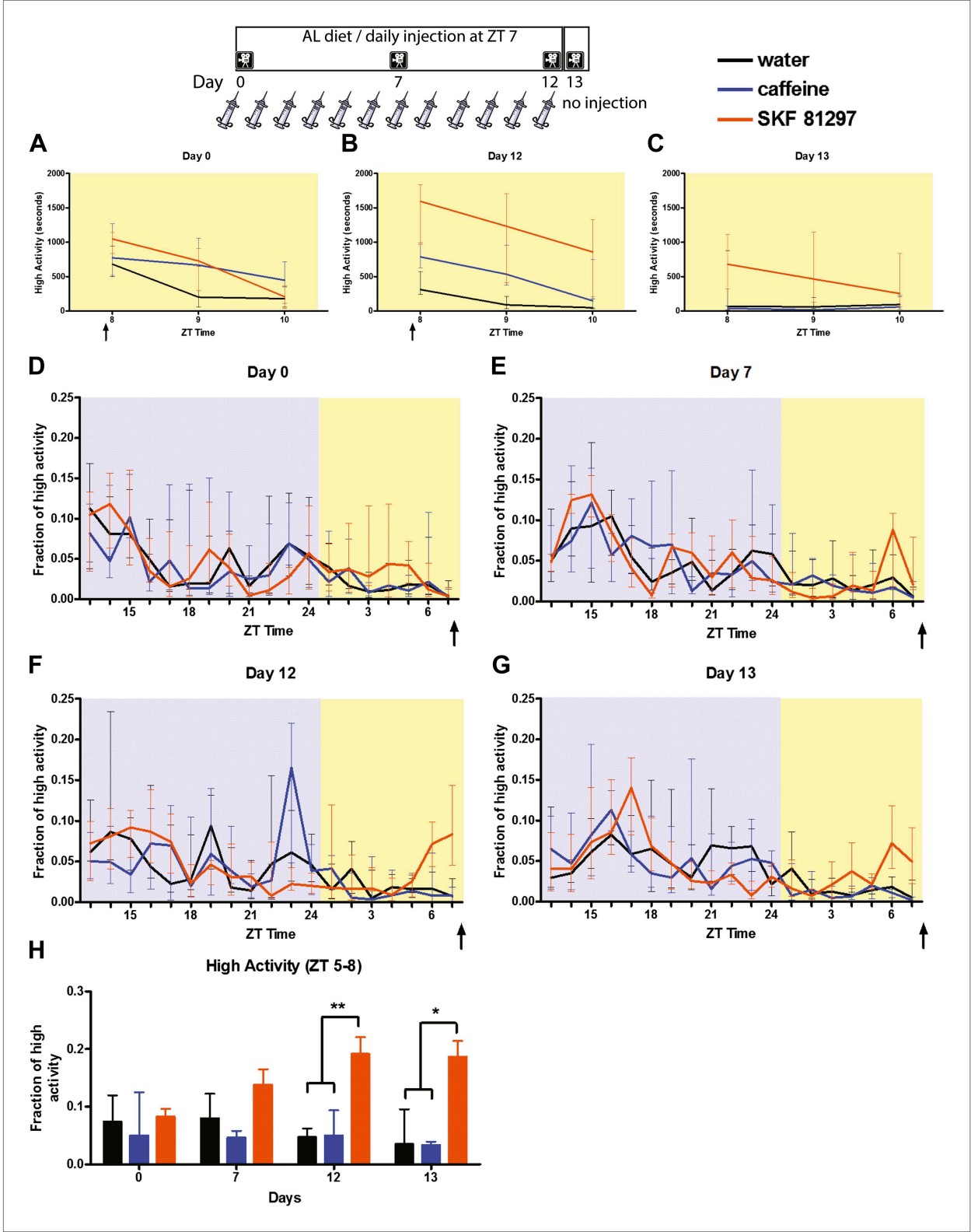

Figure 12. Mice were injected i.p. daily with water (n = 9), caffeine (n = 8–9), or SKF-81297 (n = 9–10). (A) Seconds of high activity behavior in the 2 hr after injection on day 0, the first day of injection, (B) day 12, and (C) on day 13 when no injection was performed. (D) Fraction of high activity plotted in 1 hr bins after the first day of injection, (E) seventh, (F) 12th day of injection, and (G) 1 day after the last injection. (H) The sum of normalized high activity in the 3 hr preceding scheduled injection at each behavioral measurement. * indicates p < 0.05, **p < 0.01, Mann–Whitney.

*Figure 12. Continued on next page*

*Figure 12. Continued*

The following figure supplement is available for figure 12:

**Figure supplement 1**. Mice were injected i.p. with either SKF-81297 (n = 8–12) or water (n = 8–12) for 14 days and deprived of food each day for 4 hr post-injection to prevent any drug-induced food consumption.

motivation, but that this effect is masked by CR. We speculate that the activity phenotype in D1R KO mice reflects an additional metabolic factor. Although D1R KO mice, like WT mice, exhibited hyperactivity during acute food deprivation they were significantly less active throughout the day and night during the chronic CR schedules. This was not due to metabolic collapse, given that during chronic restriction, D1R KO mice maintained nocturnal body temperatures relative to WT mice, which were hypothermic at night and early in the light period. These results suggest a different strategy used by WT and KO mice to maintain metabolic homeostasis during CR. The WT mice may conserve energy by reducing metabolism at night and expend energy by raising body temperature and locomotor activity in anticipation of a regular daily meal time. The KO mice, by contrast, switch from an initial hyperactivity to a long-term hypoactivity of sufficient magnitude to support a more normal body temperature despite restricted calorie intake. This strategy results in low levels of activity both at night and prior to meal time. Nonetheless, the reduction in FAA is significant even when normalized against total daily activity. Thus, the food anticipation phenotype in CR D1R KO mice may be explained by an alteration in the strategy to maintain metabolic homeostasis, which damps activity non-specifically throughout the 24-hr cycle and an impairment in circadian entrainment and/or incentive motivation processes that results in a disproportionate reduction of activity in anticipation of meal time.

Another notable feature of the food entrainment phenotype in D1R KO mice is that despite the marked attenuation of FAA, the preprandial rise in body temperature is essentially normal (*Figure 9*). Feeding schedules entrain circadian oscillators in many body tissues and brain regions. There is at present no direct evidence for a master food-entrainable pacemaker analogous to the retinorecipient light-entrainable pacemaker in the SCN, which in mammals is indispensible for entrainment to light:dark cycles. Indeed hypophysectomy eliminates preprandial temperature increases but not FAA (*Davidson and Stephan, 1999*), a dissociation that we have also observed with mice showing temperature entrainment without FAA (*Gallardo et al., 2012*) and mice showing FAA without showing preprandial temperature increases (*Luby et al., 2012*). Food entrainment may involve multiple parallel entrainment pathways, acting on a fully distributed, non-hierarchical system of circadian oscillators in local circuits, each responsible for generating food-entrained rhythms in tissue specific functions. The differential effect of the D1R KO on behavior and body temperature is consistent with this model.

Evidence that daily rhythms of food anticipatory activity may reflect entrainment of circadian oscillations in the dorsal striatum by daily reward schedules and D1R signaling suggest new insights into the processes that may induce and sustain daily repetitive or even addictive behaviors.

## Materials and methods

### Ethics statement

These experiments were approved by the institutional animal care committees of California Institute of Technology (protocol number 1567); Keck Science Department of Claremont McKenna College, Pitzer College, Scripps Colleges; California State Polytechnic University, Pomona (13.029), University of Washington, Seattle; and Simon Fraser University.

### Mouse strains and husbandry

For experiments performed at Caltech (*Figures 1, 2*, *Figure 2—figure supplements 1, 2*, *Figures 4–9*, *Figure 9—figure supplements 1, 2*) mice were maintained on a 13:11 Light:Dark cycle and their behavior was measured by computer vision of video recordings (described below). For data collected at Simon Fraser University (*Figure 3*, *Figure 3—figure supplement 1*, *Figure 11*), FAA was measured using horizontal running discs and motion sensors and mice were maintained on 12:12 Light:Dark cycles. Experiments utilizing dopamine-deficient mice with viral restorations of TH were performed at the University of Washington (*Figure 10*) with behavioral measurements using computer vision in mice maintained on 12:12 Light:Dark cycles. Pharmacological studies of dopamine receptor 1 activation

with SKF-81297 were performed at the Keck Science Department (*Figure 12*) and Cal Poly Pomona (*Figure 12—figure supplement 1*) in mice maintained on 12:12 Light:Dark cycles. The D1R KO mice used in this study (*Drago et al., 1994*) had been backcrossed on to C57BL/6J for at least eight generations. DNA was extracted from tail clips and the *Drd1* locus was examined by genotyped using the following primers: *neomycin* CACTTGTGTAGCGCCAAGTGC, *drd1* TCCTGATTAGCGTAGCATGGAC, *d1* GGTGACGATCATAATGGCTACGGG.

DA-deficient mice were bred and reared as described previously (*Zhou and Palmiter, 1995*). The DD mice were maintained by daily injections of l-DOPA until they were old enough to undergo surgery to re-establish TH expression in the dorsal striatum. Mice were maintained on l-DOPA for 2 weeks after surgery and then monitored closely when l-DOPA injections were terminated. Mice that were able to maintain at least 80% of body weight were considered 'rescued'. As conducted previously (*Darvas and Palmiter, 2011*), TH immune-stained striatal sections from DD-VR mice were analyzed with MacBiophotonics ImageJ software to measure fluorescence intensities in the dorsal striatum, ventral striatum, and also in the anterior part of the anterior commissure (aca), a structure that is always devoid of TH staining. For each mouse, fluorescence intensity values were divided by the size of the analyzed area to generate normalized fluorescence values.

D2R knockout mice on a C57BL/6J background were obtained from Jackson laboratory (stock # 003190). WT C57BL/6J mice (stock #000664) used for D1R pharmacology were purchased from Jackson labs.

## Scheduled feeding and behavioral measurements

Prior to being placed on special feeding protocols all mice were single-housed with AL access to food (Rodent Chow Type 5001; Lab Diet, St. Louis, MO) and water in standard microisolator cages. Daily food intake was measured over a 48-hr period beginning at least 3 days after single-housing. For all experiments, the mass of food provided to CR mice in any 24-hr period was equal to 60–70% of the average daily intake during AL food availability.

For all experiments, during scheduled feeding, food was provided in a restricted amount or for a restricted duration in the latter half of the daily light period. In different experiments, meal onset began at ZT 6, 8, or 9 (i.e., 6, 4, or 3 hr prior to lights-off), but was consistent for all groups and individual mice within experiments. Previous studies have shown that the timing of meal onset in the light period makes little to no difference in the duration and magnitude of food anticipatory activity, therefore the different meal times used in these experiments do not affect interpretation of the results (*Mistlberger, 1994*; *Mistlberger et al., 2012*). In the CR experiments, the amount of food provided in any 24-hr period was equal to 60–70% of the average daily intake during AL food availability.

D1R KO mice were fed their food allotment on the cage floor rather than in the wire food bin as we found that they were more likely to consume food when it was more readily accessible. For palatable meal feeding schedules, we used male mice only as female mice do not anticipate scheduled palatable meals robustly (*Hsu et al., 2010b*). HF group received a daily meal of 0.8 g of Bio-SERV (Flemington, NJ) high-fat diet, corresponding to ~33% of total caloric intake.

To assess activity, mice were video recorded for 23.5–24 hr. Dim red lighting was provided during the 11 hr dark cycle with red LED lights (LEDwholesalers.com, Hayward, CA) to allow acceptable contrast when recording during the night cycle. Video-based activity data were analyzed using HomeCageScan 3.0 (Clever Systems, Inc; Reston, VA); behavioral definitions were as described previously (*Steele et al., 2007*; *Hsu et al., 2010a*). High intensity activity was defined as walking, jumping, rearing, and hanging behaviors. Data were normalized by dividing the number of seconds per hour of a high activity behavior (e.g., hanging, jumping, rearing, and walking) by the total number of seconds engaged in that behavior across the ~24 hr video recording. For *Figure 3* only, data were collecting using running wheels and motion detectors as described previously (*Smit et al., 2013*). Behavioral data were exported from HomeCageScan 3.0 as excel files, which were analyzed using MATLAB programs to sum, average, and visualize data. Statistical tests were performed using GraphPad InStat, and graphs were produced using GraphPad Prism. For comparisons of behavioral data, we used non-parametric analysis: Mann–Whitney for comparisons of two groups and Kruskal–Wallis ANOVA for >2 groups. Sample sizes for each experiment are indicated in the figure legends.

For mice in *Figure 3* (Simon Fraser University), powdered food was provided. Following the baseline activity measurement period, food was presented at ZT 6 for 8 hr on the first RF day, with gradually decreasing windows of food access until stable weight was maintained on a 4-hr window of food

access. Food under this RF procedure was 20% corn oil by weight mixed in with normal powdered chow. This medium high caloric formulation was chosen to help D1R KO mice maintain healthy weight under RF conditions. Data collection under these conditions continued for 30 days. FAA was defined as locomotor activity in the 4-hr window preceding onset of food availability. Both raw FAA counts and FAA ratios (the ratio of activity in the FAA window to total activity in the day) were analyzed. Average 24 hr waveforms were constructed for each RF condition for comparisons of activity profiles including onset times, peak FAA activity, and nocturnal activity profiles.

### Body temperature experiments

For testing whether external temperature would modulate FAA in D1R KO mice, we housed D1R KO and WT mice at 30° for 1 week before beginning a 60% CR feeding schedule. Activity was measured weekly during this experiment.

For temperature monitoring experiments, mice were implanted with Mini-Mitters (Starr Life Science Corp., Oakmont, PA) subcutaneously. Mice were anesthetized using isoflurane gas. Post-operative monitoring and recovery time was provided for at least 7 days before putting mice on CR feeding schedules. Body temperature, but not activity levels, was measured in these mice.

### Pharmacology experiment

SKF-81297 and caffeine were purchased from Sigma (St. Louis, MO) (*Figure 12*) or Tocris (Bristol, United Kingdom) (*Figure 12—figure supplement 1*). Both were dissolved in water and filter sterilized prior to being injected i.p. daily for 12 days. Caffeine was injected at a dose of 15 mg/kg and SKF-81297 was injected at 5 mg/kg. Total injection volumes were approximately 250 µl per mouse. For the mice in *Figure 12—figure supplement 1*, we decreased the dose of SKF-81297 to 3 mg/kg due to seizures induced by the drug. c-Fos staining was done as described previously (*Gallardo et al., 2014*).

### Quantitative reverse-transcriptase PCR

Mice were sacrificed by sodium pentobarbital injection immediately prior to tissue collection at ZT 18, ZT 0, ZT 6, or ZT 12. Tissue was rapidly extracted, frozen on dry ice, and stored at −80°C until RNA extraction. Tissue samples were later mechanically homogenized and RNA was extracted using Trizol Reagent (Invitrogen) according to manufacturer's specifications. RNA concentrations were obtained using a Qubit 2.0 Fluorometer (Life Technologies, Carlsbad, CA) and concentrations were adjusted to 50 ng/µl. cDNA was synthesized from 500 ng RNA using High Capacity Reverse Transcription Kit (Life Technologies). Real-time PCR was performed using 2 µl cDNA with SYBR Green FastMix (Quanta Biosciences, Gaithersburg, MD) in a Step-One real time PCR system (Life Technologies). Gene expression was normalized to total RNA input. Primers used for PCR were as follows: *Per2 forward: ACCTCCCTGCAGACAAGAA, Per2 reverse: CTCATTAGCCTTCACCTGCTT.*

## Acknowledgements

We are grateful to member of the Palmiter laboratory, Teagan Wall, Henry Lester, David Anderson, and Christof Koch for support and advice.

## Additional information

### Competing interests

RDP: Reviewing editor, *eLife.* The other authors declare that no competing interests exist.

### Funding

| Funder | Grant reference number | Author |
|---|---|---|
| Broad Fellows Program in Brain Circuitry | | Andrew D Steele |
| Ellison Medical Foundation | | Andrew D Steele |
| Howard Hughes Medical Institute | | Richard D Palmiter |
| Klarman Family Foundation | | Andrew D Steele |
| Natural Sciences and Engineering Research Council of Canada | | Ralph E Mistlberger |

| Funder | Grant reference number | Author |
|---|---|---|
| National Institutes of Health | P50 NS062684 | Richard D Palmiter |
| Science Educational Enhancement Services Cal Poly Pomona | | Antonio Aguayo |
| Natural Sciences and Engineering Research Council of Canada | | Mateusz Michalik, Danica F Patton |
| Claremont McKenna Interdisciplinary Science Scholarship | | Chris H Chang |
| Howard Hughes Medical Institute | Summer Fellowship | Emily E Meyer |

The funders had no role in study design, data collection and interpretation, or the decision to submit the work for publication.

## Author contributions

CMG, ADS, Conception and design, Acquisition of data, Analysis and interpretation of data, Drafting or revising the article; MD, MM, Conception and design, Acquisition of data, Analysis and interpretation of data; MO, SAS, Acquisition of data, Analysis and interpretation of data, Drafting or revising the article; CHC, TFH, EEM, AA, EMH, KK, JI, MP, ANS, DFP, Acquisition of data, Analysis and interpretation of data; JSM, Acquisition of data, Drafting or revising the article; REM, RDP, Conception and design, Analysis and interpretation of data, Drafting or revising the article

## Ethics

Animal experimentation: This study was performed in accordance with the recommendations in the Guide for the Care and Use of Laboratory Animals of the National Institutes of Health. All of the animals were handled according to approved institutional animal care and use committee (IACUC) protocol 1567.

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
