## [Decision Letter]

Thank you for sending your work entitled “Dopamine receptor 1 neurons in the dorsal striatum regulate food anticipatory circadian activity rhythms in mice” for consideration at *eLife.* Your article has been favorably evaluated by a Senior editor and 3 reviewers, one of whom, Leslie Griffith, is a member of our Board of Reviewing Editors.

The Reviewing editor and the other reviewers discussed their comments before we reached this decision, and the Reviewing editor has assembled the following comments to help you prepare a revised submission.

The reviewers found this paper to be a potentially important contribution to our understanding of FAA. There were, however, a number of substantial and minor concerns that need to be dealt with before this paper would be suitable for publication. The full reviews are appended below. The major issues, requiring re-review, are the following:

1) There are numerous apparent inconsistencies in the timing of the meals and the behavioral assessments and this compromises the clarity of the presentation. These inconsistencies prevent the reader from evaluating several of the key findings adequately (see Reviewer #2 and Reviewer 3# points 1-4). This is a crucial shortcoming that would need to be addressed for the paper to be reassessed by the reviewers.

2) The specificity of the viral manipulation needs to be established in a quantitative manner (see Reviewer #2 comments below).

3) There were no data on the pattern of Per2 expression in ad lib feeding. These data are important the authors' claims and need to be shown (see Reviewer #1 and Reviewer #3 point 5).

Further detail is provided below:

*Reviewer #1*:

This paper is a very thorough investigation of the nature of the circadian processes underlying food anticipation in mice. This phenomenon has been investigated for many years and has been a very difficult to understand mechanistically due to the fact that there are many extrinsic and intrinsic variables that animals integrate in making behavioral decisions. The authors do an excellent job of investigating what are probably the most important variables and come up with a very convincing story about DA in the striatum being the driver of the relevant peripheral oscillator. I think that this paper, because it answers a very long-standing question in a rigorous manner is an important contribution.

Minor comments:

1) The fact that there is a basal Per2 rhythm with peak at ZT23 is kind of buried. It is relevant that there is a 12h phase shift with CR entrainment and the authors might want to make this clearer.

2) In some figures (e.g. Figure 4 vs D) the Y axes are different on panels that the reader might want to compare. If you normalize the axes, the cross panel contrasts would be more obvious.

*Reviewer #2*:

This study by Gallardo and colleagues explores the role of DA in food anticipatory activity (FAA) in mice. These daily rhythms of FAA are of broad interest to the community as prior work has shown that the SCN is not involved. Prior work has indicated a role for DA in FAA still much is not known about the underlying mechanisms. In this study, the authors show that D1R KO mice do not show much FAA while the D2R KO mice have normal FAA. In an important advance, the authors report that use of a viral construct to express DA in the dorsal striatum of dopamine-deficient mice allows these mice to develop FAA. Finally, the administration of D1 agonist itself is sufficient to establish anticipatory activity in wild-type mice.

There are several strengths of this work: The topic is of interest, the writing is clear, and citations appropriate; The Introduction nicely sets up the problem; Some of the data are very clear including the impact of the loss of D1R and the rescue of FAA in dopamine deficient mice by injection of the virus in the dorsal striatum.

Still there are some weaknesses:

The Results need to be pruned. The authors present all of the data with equal weighting while there are clearly some experiments that are critical while others are more of controls.

The issue of the anatomical specificity of the virus-driven DA expression is critical. The authors want to claim that the dorsal striatum is important for this behavioral rescue but they do not present enough data for us to evaluate this claim. How many mice were injected? How was the expression of the DA quantified? They show us a nice picture but we need more information if we are to believe that the dorsal striatum is critical.

*Reviewer #3*:

The manuscript reports a suppression of food anticipatory activity (FAA) in dopamine D1 receptor, but not D2 receptor, knockout (KO) mice. They also show that 24-hour scheduled D1R activation in the dorsal striatum leads to anticipatory-like activity in mice. The results are interesting and have potential implications for a wide range of biological processes. The authors also performed several different types of control and follow-up experiments, providing nuanced detail about the nature of FAA in the D1R KO mice. There are, however, several elements of the manuscript that raise questions and prevent the reader from drawing clear conclusions as to the role of D1 receptors in FAA. These and other concerns about the manuscript are detailed below.

1) The feeding times for the food restriction vary throughout the manuscript. This makes it difficult to accurately compare the data from figure to figure/experiment to experiment. Further, while quantitative analyses are said to be performed on the 3h prior to feeding, the figure labels consistently refer to a 2h interval (e.g., ZT5-7 in Figure 1). Additionally, the feeding time is not accurately reported for each set of data. For example, Figure 2 appears to show the meal occurring at ZT8 (which is what is reported for this experiment in the text) and then Figure 2 shows activity being score for ZT5-7 (a 2h interval), but Figure 2 supplement shows a meal at ZT8, with activity being scored ZT7-9 (Figure 2—figure supplement 2), and the figure legend states that these data are from the same mice as Figure 2. These details need to be corrected and clarified throughout the manuscript so that the reader can make clear comparisons between the figures presented.

2) The data presented in the figures, with the exception of Figure 3, are said to be median values. This is an unusual way to report the average. Do the average waveforms also use median values, or only the histograms? It is conventional to report mean value, and doing so (or reporting median values for Figure 3) would allow the reader to compare data across figures of this manuscript.

3) It is difficult to tell whether the D1R knockout mice completely lack FAA or whether it is merely attenuated. In Figure 3 (wheel-running), the mice appear to be showing FAA, but the “high activity” measure does not appear to capture this difference. Statistics are reported for differences between D1R and WT mice, but were within-subjects analyses done to determine whether WT and D1R mice were showing increased activity across time on restricted feeding? Please include these analyses.

4) The presentation of the activity waveform figures in all but Figure 3 makes it hard to view the FAA. Shifting the x-axis so that the meal time is closer to the center would allow the reader to view the data more easily. I would suggest making the x-axis for all activity figures match that of Figure 3.

Additionally, the authors should include actograms (the original data) presenting the data for representative D1R KO and WT mice on restricted feeding. Because none of the actograms are shown, it is difficult to assess the quality of the activity records. (This is analogous to showing only the quantitated values from Western blot experiments without showing examples of the original gel images in biochemical experiments.) These actograms could go into the supplement if absolutely necessary, but it would be preferable to include representative examples in the main figures.

5) The Per2 expression data presented in Figure 11 are not interpretable without a similar time course of (control) data from animals fed ad lib.

6) There is no information in the figure legend for Figure 1.

[Editors' note: further revisions were requested prior to acceptance, as described below.]

Thank you for resubmitting your work entitled “Dopamine receptor 1 neurons in the dorsal striatum regulate food anticipatory circadian activity rhythms in mice” for further consideration at *eLife.* Your revised article has been favorably evaluated by a Senior editor and Leslie Griffith of the Board of Reviewing Editors. The manuscript has been improved but there are some remaining issues that need to be addressed before acceptance, as outlined below:

This revision has improved an already interesting manuscript. That said, there are still a few major concerns that have not been properly addressed.

1) Individual animal data. Authors only included individual records for day 21 of RF (Figure 2—figure supplement 2); there are no data from prior to RF. In the main figure, they present summary data for days -7, 0, 7,14, 21, & 28. It would be nice to see at least the -7 (prior to RF) data for comparison, preferably from the same animals already shown in F2S2 (so we can actually see how an individual animal responds following RF).

2) Stats on FAA. The authors explain clearly the differences in analysis methods in the rebuttal, but they still either need to put in a statistical treatment of the data (showing WT animals had FAA and whether or not it was significant in D1RKO animals) or scale back claims. Either is acceptable.

3) The confusion about ZT7-9. The wave form and starting times for the recordings are the result of the authors' unusual way of assessing behavior for only one day at a time by moving the animal into the video apparatus. The confusion is occurring because ZT5-7 is a 2h period ending at ZT7. If data are scored up until ZT8, this should be written as “ZT5-8.” ZT (zeitgeber time) is a form of time, and thus a continuous variable (not an integer). ZT5-7 is, by definition, 2h-ZT5:00-ZT7:00. Writing ZT5-7 when you intend to include all of hour 7 is the equivalent of saying an event occurs from 5:00am-7:00am when you mean it occurs from 5:00am to 8:00am. All of the figures and text should be updated to denote a 3h period. This notation of times must be corrected in the manuscript before acceptance of the paper.

The authors need to make explicit all the various differences in the experiments (sites, labs, equipment), and then they can use this as a 'strength' as they argue in their rebuttal. We recommend that they include a section in the Methods section on these issues.

---

## [Author Response]

*1) There are numerous apparent inconsistencies in the timing of the meals and the behavioral assessments and this compromises the clarity of the presentation. These inconsistencies prevent the reader from evaluating several of the key findings adequately (see Reviewer #2 and Reviewer 3# points 1-4). This is a crucial shortcoming that would need to be addressed for the paper to be reassessed by the reviewers*.

There are two reasons for inconsistencies in meal time presentation: 1) a mistake on our part in Figure 2—figure supplement 2 and Figure 6 (zt 5-7, not 7-9). We apologize for this error and have made the appropriate corrections in the revised manuscript. 2) The other reason for differences in feeding times was a result of the experiments being conducted over a 4+ year period in 5 separate laboratory settings (Caltech, Simon Fraser, Keck Science Department, UWash, and Cal Poly Pomona). For example, at some institutions the light:dark cycle was 13:11 and at others 12:12. For FAA studies mice were fed consistently during the middle-to-late portion of the light cycle and the range was different by only 3 hours; all mice were fed between ZT6 and ZT8 for our study. A great many restricted daytime feeding experiments have been done over the past ∼40 years, and these studies show that the parameters of food anticipatory activity (most importantly, its duration and amplitude) vary little or not at all with the time within the light period when the mealtime is initiated. In our experiments the mealtimes are consistent within experiments, which is the only methodological factor that is critical for interpreting differences between WT and KO mice. In fact, the variability of mealtimes across experiments can be viewed as a strength of the study, because we consistently observed the deficit in food anticipatory activity in caloric restriction experiments regardless of when the daily meal started in a particular experiment.

With respect for evaluation of behavior, we used several different methods (computer vision, running wheel, and motion sensors) to thoroughly characterize the D1R knockout mice and their deficit in FAA. *Prima face* this can be confusing, but it is actually a great strength of our study. To ensure that our results are reproducible across laboratories we tested KO mice in different settings with different techniques. The lack of FAA observed in D1R KO mice using computer vision was more pronounced than what was observed at Simon Fraser University using disc running wheels and motion sensors, and continuous daily recordings. Nonetheless, the defect was still significant at SFU. We replicated the D1R KO FAA study using computer vision data at Keck Science Department but did not present in our manuscript because the study was terminated at day 14 (Figure 13):Author response image 1.replication of D1R KO CR phenotype at Keck Science Department using computer vision to measure activity levels. n=7-10 mice per group.

In summary, across several laboratory settings with slight variations in feeding times, we observed a diminution or in some cases a complete lack of FAA in D1R KO mice. Further, detailed responses to the queries of reviewers 2 and 3 can be found below.

*2) The specificity of the viral manipulation needs to be established in a quantitative manner (see Reviewer #2)*.

This model relies primarily on a behavioral “rescue” phenotype (ability to survive without daily L-DOPA injections). We did perform a histological assessment of n=5 of the 7 dopamine deficient mice with restorations of tyrosine hydroxylase expression to the dorsal striatum and we have added a panel to Figure 10 (Figure 10).

*3) There were no data on the pattern of Per2 expression in ad lib feeding. These data are important the authors' claims and need to be shown (see Reviewer #1 and Reviewer #3 point 5)*.

The data that we have available to report at the present time confirm that there is a daily rhythm of Per2 expression in the dorsal striatum of WT mice anticipating a daytime feeding opportunity, and show for the first time that this rhythm is severely attenuated (statistically absent) in D1R KO mice. These data permit a substantive conclusion that dopamine signaling at D1 receptors is necessary for a clock gene rhythm in the dorsal striatum of food-restricted mice. This conclusion does not require data on WT and KO mice on ad-lib food access. Ad-lib groups are only required if we were trying to make a novel claim that clock gene expression in the dorsal striatum is shifted by restricted daytime feeding schedules. That claim has already been made in the literature by several groups (dating back to [62]). While we do cite prior studies to suggest that the Per2 rhythm in the WT mice is shifted by comparison with ad-lib fed WT mice in previous studies, that is not the important point; rather, it is the loss of this rhythm in KO mice that is the critical, novel finding.

Reviewer #1:

*1) The fact that there is a basal Per2 rhythm with peak at ZT23 is kind of buried. It is relevant that there is a 12h phase shift with CR entrainment and the authors might want to make this clearer*.

The Results section here is worded as follows:

“WT mice exhibited a 24-hr rhythm with peak expression at ZT12 (lights-off), which is advanced by comparison with the daily rhythm in AL fed rodents (e.g., (16)). KO mice exhibited a markedly attenuated daily rhythm of striatal *Per2* expression, due primarily to reduced expression at ZT12 (Figure 11).”

We believe that this is a clear statement of the results, and are not sure in what way it should be improved. Our goal was to compare WT and KO mice on restricted feeding, and thus while we do note that the timing of the rhythm in the WT group is consistent with a phase shift, that is a secondary point (as this has already been demonstrated in several prior studies).

*2) In some figures (e.g.*
Figure 4
*vs D) the Y axes are different on panels that the reader might want to compare. If you normalize the axes, the cross panel contrasts would be more obvious*.

Corrected in Figures 2, 4, 8 and 10.

Reviewer #2*:*

*[…] There are several strengths of this work: The topic is of interest, the writing is clear, and citations appropriate; The Introduction nicely sets up the problem; Some of the data are very clear including the impact of the loss of D1R and the rescue of FAA in dopamine deficient mice by injection of the virus in the dorsal striatum*.

*Still there are some weaknesses*:

*The Results need to be pruned. The authors present all of the data with equal weighting while there are clearly some experiments that are critical while others are more of controls*.

We thank the reviewer for their appreciation of our work. With respect to the equal weighting of the results, this is certainly the case in the Results section and derives from 1) author style and our belief that these controls were vitally important to demonstrate that the D1R knockout FAA phenotype was not an epiphenomenon 2) the discouragement of supplemental figures by *eLife*. If this were, say, a *Nature* or *Science*, then Figures 3, 4, 5, 6, 7 and 9 would all have been relegated to a supplement. The Discussion section does a better job at emphasizing the importance of particular results. We are open to specific suggestions as to how to decrease weight of particular experiments but do wish to avoid creating additional supplemental figures.

*The issue of the anatomical specificity of the virus-driven DA expression is critical. The authors want to claim that the dorsal striatum is important for this behavioral rescue but they do not present enough data for us to evaluate this claim. How many mice were injected? How was the expression of the DA quantified? They show us a nice picture but we need more information if we are to believe that the dorsal striatum is critical*.

The quantitation added to the manuscript addresses this very valid concern (see Figure 2 above). N=7 dopamine deficient, virally rescued mice were used in our study.

Reviewer #3:

*[…] There are, however, several elements of the manuscript that raise questions and prevent the reader from drawing clear conclusions as to the role of D1 receptors in FAA. These and other concerns about the manuscript are detailed below*.

*1) The feeding times for the food restriction vary throughout the manuscript. This makes it difficult to accurately compare the data from figure to figure/experiment to experiment. Further, while quantitative analyses are said to be performed on the 3h prior to feeding, the figure labels consistently refer to a 2h interval (e.g., ZT5-7 in*
Figure 1*). Additionally, the feeding time is not accurately reported for each set of data. For example,*
Figure 2
*appears to show the meal occurring at ZT8 (which is what is reported for this experiment in the text) and then*
Figure 2
*shows activity being score for ZT5-7 (a 2h interval), but*
Figure 2
*supplement shows a meal at ZT8, with activity being scored ZT7-9 (*Figure 2—figure supplement 2*), and the figure legend states that these data are from the same mice as*
Figure 2*. These details need to be corrected and clarified throughout the manuscript so that the reader can make clear comparisons between the figures presented*.

Please see response #1 to the major issues for an explanation of feeding times. We thank the reviewer for catching the mistake in Figure 2’s supplement (Figure 2—figure supplement 1). To clarify, for all our experiments, a window of 3 hr prior to feeding time was used. For example, ZT 5-7 is inclusive of hours, 5, 6, and 7 and therefore represents the 3 hr prior to feeding. It is the sum of activity in each of the three hours prior to the feeding time, which occurs at ZT 8. In the first paragraph of the Results section we had stated our definition as follows: “In both D2R KO and WT mice fed 60% CR daily, we observed similar acquisition and maintenance of FAA, defined as normalized high activity in the 3 hr before feeding (ZT 5-7) (Figure 1).” If that is not clear enough then we are open to suggestions on how to avoid confusion.

*2) The data presented in the figures, with the exception of*
Figure 3*, are said to be median values. This is an unusual way to report the average. Do the average waveforms also use median values, or only the histograms? It is conventional to report mean value, and doing so (or reporting median values for*
Figure 3*) would allow the reader to compare data across figures of this manuscript*.

Here we are running into different conventions for how data are presented; for data collected on home cage behaviors using computer vision we always present median +/- interquartile range (in our 6 previous FAA papers: *PLOS ONE* 5(9): e12903, *PLOS ONE* 5(11): e15429, *PLOS ONE* 6(3): e18377, *PLOS ONE* 7(5): e37992, *PLoS ONE* 7(7): e41161, *PLOS ONE* 9(5): e95990, PLOS ONE, e12903). Since our data does not fall on a normal distribution, the median is more reflective of the non-parametric statistical tests that we use. Essentially, the mean value has no meaning in terms of our statistics so it seems misrepresentative to present means +/- standard error.

For the wheel-running data presented in Figure 3 the mean +/- standard error is used as following normal convention for this lab and this type of data.

*3) It is difficult to tell whether the D1R knockout mice completely lack FAA or whether it is merely attenuated. In*
Figure 3
*(wheel-running), the mice appear to be showing FAA, but the “high activity” measure does not appear to capture this difference. Statistics are reported for differences between D1R and WT mice, but were within-subjects analyses done to determine whether WT and D1R mice were showing increased activity across time on restricted feeding? Please include these analyses*.

The lack of FAA is more apparent when measured by computer vision than with disc-running behavior. This may reflect in part the fact that in the disc running experiments, data collection was continuous (as per convention with this measure), whereas the computer vision analyses (more computationally intense) were done on ∼1 day of 24h video recording/week. There are some D1R KO mice that completely fail to increase activity (measured by computer vision) for meal time but most mice do show a small increase in activity before meal time. The individual mouse data added to Figure 2—figure supplement 2 and Figure 3—figure supplement 1 should help to clarify this and show the mouse-to-mouse variation. Within mice, after the first 1-2 weeks, FAA was typically stable across time, as illustrated in Figure 3—figure supplement 1.

*4) The presentation of the activity waveform figures in all but*
Figure 3
*makes it hard to view the FAA. Shifting the x-axis so that the meal time is closer to the center would allow the reader to view the data more easily. I would suggest making the x-axis for all activity figures match that of*
Figure 3.

*Additionally, the authors should include actograms (the original data) presenting the data for representative D1R KO and WT mice on restricted feeding. Because none of the actograms are shown, it is difficult to assess the quality of the activity records. (This is analogous to showing only the quantitated values from Western blot experiments without showing examples of the original gel images in biochemical experiments.) These actograms could go into the supplement if absolutely necessary, but it would be preferable to include representative examples in the main figures*.

The x-axes are aligned to start with the beginning of video recording, which has a substantial impact on activity as the mice become very active for the 1-2 hr after we put them into the video recording room and a new cage. Thus, it would be confusing to replot the data as suggested because there would be a spike in activity at ZT 8, when the video recording was initiated, that is often as high or higher than the FAA peak.

Data from individual mice have been added as supplements to Figures 2 and 3.

*5) The Per2 expression data presented in*
Figure 11
*are not interpretable without a similar time course of (control) data from animals fed ad lib*.

See explanation above in response to major concern #3.

*6) There is no information in the figure legend for*
Figure 1.

We have added the text “(F) The fraction high activity in the 3 hr before feeding time (ZT 5-7) for mice on *ad libitum* diets.”

Finally, we also added one additional experiment to strengthen the dopamine receptor agonist data, adding a supplemental figure to Figure 12 (Figure 12—figure supplement 1). This figure replicates the data presented in the main text with an additional control measure: mice were not fed for 4 hr post-injection with drug to rule out the confounding factor that D1R agonist may increase feeding behaviors and thereby schedule feeding and entrain behavior due to the scheduled feeding. We also performed c-Fos staining to confirm that the drug was inducing activation of neurons in wild-type but not in D1R KO mice.

[Editors' note: further revisions were requested prior to acceptance, as described below.]

*[*…*] This revision has improved an already interesting manuscript. That said, there are still a few major concerns that have not been properly addressed*.

*1) Individual animal data. Authors only included individual records for day 21 of RF (*Figure 2—figure supplement 2*); there are no data from prior to RF. In the main figure, they present summary data for days -7, 0, 7,14, 21, & 28. It would be nice to see at least the -7 (prior to RF) data for comparison, preferably from the same animals already shown in F2S2 (so we can actually see how an individual animal responds following RF)*.

We have added an additional supplemental figure to show 6 individual mice on day 0, the first day of scheduled feeding, and 6 individual mice day 21. We chose day 0 because day -7 is what we consider the “habituation” day as it is the first time the mice have been transported to the video recording room and placed in front of the cameras. In Figure 2—figure supplement 2, randomly selected wild-type mice are shown and in Figure 2—figure supplement 3, 6 randomly selected D1R KO mice are shown. While it is difficult to quantitatively define FAA (in the past we have used the ratio of night time activity peak to FAA peak greater than 0.5), it appears that 2 out of 6 D1R KO mice show some FAA whereas all WT mice show strong FAA on day 21. We do not intend to argue that D1R KO mice have no FAA just that it is attenuated and sometimes absent.

*2) Stats on FAA. The authors explain clearly the differences in analysis methods in the rebuttal, but they still either need to put in a statistical treatment of the data (showing WT animals had FAA and whether or not it was significant in D1RKO animals) or scale back claims. Either is acceptable*.

Our analysis up to this point had concerned whether the amount of normalized FAA at individual time points was less in D1R KO mice compared to WT controls. But the reviewer brings up an interesting point, which is to ask the question of “do WT and D1R KO mice develop FAA over time?” Such a within group comparison had not been performed. To that end we performed a non-parametric ANOVA, Kruskal-Wallis test with post-test for the mice that were part of the computer vision study at Caltech comparing FAA on days 0, 7, 14, 21, and 28 for WT or D1R KO mice. We found that for WT mice p < 0.0001 and the post-test showed that the difference between day 0 and 7 was not significant (p>0.05) but the difference between day 0 and 14 was significant (p<0.01), as was the differences between day 0 and day 21 (p<0.001) and day 0 and day 28 (p<0.001). For D1R KO mice P = 0.3145, indicating that there was not a significant difference in FAA over the time points measured.

For the running wheel/motion sensor study conducted at SFU, we performed statistical testing looking at each mouse. We added the following text:

“In only one KO mice were the FAA counts and the FAA ratios significantly increased during the last 10 days of restricted feeding compared to the last 10 days of ad-lib food access (counts: paired t(9)=6.52, p<.001; ratios t(9)=5.22, p=.005), whereas the counts and ratios were significantly increased in all of the WT mice at p<.00001.”

*3) The confusion about ZT7-9. The wave form and starting times for the recordings are the result of the authors' unusual way of assessing behavior for only one day at a time by moving the animal into the video apparatus. The confusion is occurring because ZT5-7 is a 2h period ending at ZT7. If data are scored up until ZT8, this should be written as “ZT5-8.” ZT (zeitgeber time) is a form of time, and thus a continuous variable (not an integer). ZT5-7 is, by definition, 2h-ZT5:00-ZT7:00. Writing ZT5-7 when you intend to include all of hour 7 is the equivalent of saying an event occurs from 5:00am-7:00am when you mean it occurs from 5:00am to 8:00am. All of the figures and text should be updated to denote a 3h period. This notation of times must be corrected in the manuscript before acceptance of the paper*.

We have corrected the figures and associated text to more clearly indicate the 3-hour time bin that we have chosen for FAA.

*The authors need to make explicit all the various differences in the experiments (sites, labs, equipment), and then they can use this as a 'strength' as they argue in their rebuttal. We recommend that they include a section in the Methods section on these issues*.

Rather than created an additional section into the Materials and methods section we further clarified the site of teach experiment in relation to each figure by including the following text in the “mouse strains and husbandry” subsection:

“For experiments performed at Caltech (Figures 1, 2, 4, 5, 6, 7, 8 and 9) mice were maintained on a 13:11 L:D cycle and their behavior was measured by computer vision of video recordings [described below]. For data collected at Simon Fraser University (Figures 3 and 11), FAA was measured using horizontal running discs and motion sensors and mice were maintained on 12:12 L:D cycles. Experiments utilizing dopamine deficient mice with viral restorations of TH were performed at the University of Washington (Figure 10) with behavioral measurements using computer vision in mice maintained on 12:12 L:D cycles. Pharmacological studies of dopamine receptor 1 activation with SKF-81297 were performed at the Keck Science Department (Figure 12) and Cal Poly Pomona (Figure 12—figure supplement 1) in mice maintained on 12:12 L:D cycles.”